# When is an Embedder More Promising than Another?

[†]**Maxime** D[ARRIN][1,2,3,4]    [†]**Philippe** F[ORMONT][1,2,4,5]    **Ismail** B[EN] A[YED][1,5]
**Jackie Chi Kit** C[HEUNG][2,3]    **Pablo** P[IANTANIDA][1,2,4,6]
[1]International Laboratory on Learning Systems, [2]Mila - Quebec AI Institute, [3]McGill University
[4]Université Paris-Saclay, [5]ÉTS Montréal, [6]CNRS, CentraleSupélec, [†]equal contribution
`maxime.darrin@mila.quebec, philippe.formont@mila.quebec`

## Abstract

Embedders play a central role in machine learning, projecting any object into numerical representations that can, in turn, be leveraged to perform various downstream tasks. The evaluation of embedding models typically depends on domain-specific empirical approaches utilizing downstream tasks, primarily because of the lack of a standardized framework for comparison. However, acquiring adequately large and representative datasets for conducting these assessments is not always viable and can prove to be prohibitively expensive and time-consuming. In this paper, we present a unified approach to evaluate embedders. First, we establish theoretical foundations for comparing embedding models, drawing upon the concepts of sufficiency and informativeness. We then leverage these concepts to devise a tractable comparison criterion (information sufficiency), leading to a task-agnostic and self-supervised ranking procedure. We demonstrate experimentally that our approach aligns closely with the capability of embedding models to facilitate various downstream tasks in both natural language processing and molecular biology. This effectively offers practitioners a valuable tool for prioritizing model trials.[1]

## 1 Introduction

Embeddings are a prominent tool in machine learning and are used in multiple fields, such as natural language processing [64, 83], computer vision [93, 59, 12, 53] or bioinformatics [67, 3, 23, 112]. These models embed objects such as images, texts, or molecules into numerical representations that can be used to perform numerous downstream tasks by preserving key features of the object [76, 111].

Depending on the data modalities, intended purpose, and available resources, embedders showcase a wide variety of architectures, training settings (unsupervised, supervised, self-supervised, etc.), objectives (masked language modeling, contrastive learning, etc.) [20, 100, 78, 112, 39], and datasets [65, 86, 31, 38, 5, 117]. And more recently, foundation models have become a natural starting point to create embedders [21, 106, 52, 73].

This diversity and variety of options makes selecting the most promising embedders for a data distribution challenging [75]. Most work evaluates embedders focusing on the performance they enable on a finite set of downstream tasks [85, 17, 90, 91, 81, 24]. Nevertheless, this evaluation process encounters two primary limitations. Firstly, it is **not scalable** concerning the number of embedders and tasks, as it requires fitting a downstream model for each task. Hence, prioritizing the evaluation of the most promising models becomes essential to mitigate computational costs. Secondly, **acquiring high-quality labels** can be a **time-consuming** and notably **expensive** endeavor in various applications. To overcome these limitations, in this paper, we explore task-agnostic evaluation metrics

---

[1]The code used to perform all experiments is available at `https://github.com/ills-montreal/emir`

38th Conference on Neural Information Processing Systems (NeurIPS 2024).

for embedders relying solely on pairwise comparisons between embedders, i.e., without the need for labeled data in downstream tasks.

More specifically, our contributions can be summarized as follows:

1. **An innovative theoretical framework for comparing embedding models:** We cast the problem of ranking embedders into the noisy communication channels ordering (Sec. 2.2) and statistical experiments comparison settings (Sec. 2.3). We exploit the notions of sufficiency and informativeness and relax them, leveraging the concept of deficiency introduced by Le Cam [63] (Sec. 2.4), which is reframed to account for concepts and features. These concepts provide us with tools to establish an embedder ranking.

2. **A practical relaxation:** Estimating deficiency presents significant challenges. We propose the concept of information sufficiency (IS), which quantifies the information required to simulate one embedder from another (Sec. 3). We estimate the information efficiency to get a task-agnostic and label-free comparison tool for embedders evaluation.

3. **Extensive experimental validation:** The expected IS correlates with the ability of embedders to enable a wide range of downstream tasks. In NLP (Sec. 5) and molecular modeling (Sec. 6), our method respectively achieves Spearman ranking correlations of 0.90 (56 tasks) and 0.94 (31 tasks); providing an efficient model trial prioritization tool for practitioners.

## 1.1 Related works

**Embedding evaluation.** Embedding evaluation is mainly performed based on a limited set of downstream tasks [22, 91, 81, 24], for which the embeddings are used as inputs to smaller models. Therefore, embedders evaluation is field- and task-specific. In NLP, [41, 85] they rely on a limited set of tasks; more recently, the Massive Text Embedding Benchmark (MTEB) [75] followed this task-oriented trend and offered standardized test bed for embedders encompassing various downstream tasks in NP. Devising statistical tests to compare models and learning algorithms has a long history [30]. However, most works propose statistical tests relying on the performance of the downstream tasks of interests [60, 11]. Other works study the expressiveness of embedders and connect it to performance on downstream tasks [107, 25], but mostly focus on geometrical properties of the high dimensional representation in self-supervised learning settings [2, 42, 45].

**Probing.** While probing methods do not aim at comparing embedders, they evaluate their representations to discover what these models have learned. They train small models on the internal representations of large models to perform specific downstream tasks. These procedures allow researchers to assess what information is present and recoverable from these embeddings [10, 1, 88, 84]. Other work proposed measuring mutual information (MI) between internal representations and labels. It has been used to evaluate the difficulty of a dataset as the predictiveness of the labels using the features [35]. For instance, [97] evaluates the utility of representations in astrophysics to predict physical properties. Following this trend, [54] leverages the point-wise MI between Gaussian distributions to evaluate text-to-images and image-to-text generative models. However, none of these methods have focused on comparing embedders in the general case to the best of our knowledge.

## 2 Theoretical Foundations for Comparing Embedding Models

### 2.1 Background and notation

We assume that all considered spaces are standard Borel [28] Each such space $\mathsf{U}$ is equipped with its Borel $\sigma$-algebra $\mathcal{B}(\mathsf{U})$. The set of all probability measures on $\mathsf{U}$ is denoted by $\mathcal{P}(\mathsf{U})$ The total variation distance between $P$ and $Q$ is denoted by $\|P - Q\|_{\text{TV}}$. Given a joint probability measure $P_{XY}$ induced by two random variables $X \in \mathsf{X}$ and $U \in \mathsf{U}$, the Mutual Information [27] is denoted by $I(X; U)$. A Markov (or transition probability) kernel between $\mathsf{X}$ and $\mathsf{U}$ is a mapping $P_{U|X} : \mathcal{B}(\mathsf{U}) \times \mathsf{X} \to [0, 1]$. The space of all such $P_{U|X}$ is denoted by $\mathcal{K}(\mathsf{U}|\mathsf{X})$ and $(M \circ P_{U|X})(V|x)$ indicates the composition of Markov kernels $M \in \mathcal{K}(\mathsf{V}|\mathsf{U})$ and $P_{U|X} \in \mathcal{K}(\mathsf{U}|\mathsf{X})$. For further details, refer to Appendix A.

### 2.2 Sufficiency and informativeness ordering of embedding models

We aim to compare embedding models without relying on labeled data for downstream tasks. Let us consider two embedding models represented by their Markov kernels (or transition probabilities)

$P_{U|X} \in \mathcal{K}(\mathsf{U}|\mathsf{X})$ and $P_{V|X} \in \mathcal{K}(\mathsf{V}|\mathsf{X})$, any target set $\mathsf{Y}$ of (discrete or continuous) concepts and feature space $\mathsf{X}$ with joint probability measure $P_{YX} \in \mathcal{P}(\mathsf{Y} \times \mathsf{X})$ induced by random variables $(Y, X) \in \mathsf{Y} \times \mathsf{X}$, as illustrated in Figure 1. First, we study the question:

**What sufficient conditions must be met by the embedding model $U$ relative to $V$ to guarantee that $I(Y;U) \geqslant I(Y;V)$ for all distributions $P_{YX}$ ?**

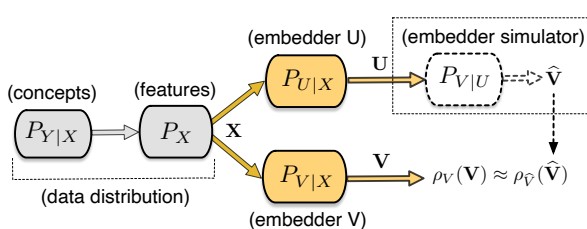

Figure 1: Communicating a concept $y \in \mathsf{Y}$ over two embedding models with prediction $\rho_V(V)$.

From an information-theoretic perspective [27], the quality of an embedding model can be likened to the capacity of a noisy communication channel with an uncoded input (e.g., a text, a molecule...), where a downstream task of interest is performed at the output (the embedding) of the channel. Let $Y \in \mathsf{Y}$ represent the message (the source) to be communicated over both channels; $X$ represents the transmitted signal; and $P_{U|X}$ and $P_{V|X}$ the communication channels with outputs $U$ and $V$, respectively. This process is illustrated in Figure 1. It naturally satisfies the Markov chain $Y \leftrightarrow X \leftrightarrow (U, V)$. A desirable property is that the embedding models $U$ and $V$ retain as much pertinent information as feasible to predict $Y$.

We shall be interested in the underlying information relationships between those embedding models that can be interpreted as channel $U$ being "more informative" for communicating $Y$ than channel $V$. The first attempt to introduce an ordering between communication channels appears in Shannon [94]. Körner and Marton later introduced [57] the concepts of "less noisy" (or more informative) and "degraded" (or sufficiency) orderings between channels.

**Definition 1** (Sufficiency and informativeness orderings [57]). Let $P_{U|X}$ and $P_{V|X}$ be two Markov kernels (embedding models).

- **Sufficiency $U \succcurlyeq_S V$.** The embedding model $P_{U|X}$ is said to be "sufficient" for the embedding model $P_{U|X}$ (or $V$ to be degraded w.r.t. $U$) if and only if there exists another Markov kernel $M \in \mathcal{T}(\mathsf{V}|\mathsf{U})$ such that $\mathbb{E}\|M \circ P_{U|X} - P_{V|X}\|_{\mathrm{TV}} = 0$, i.e. $V$ can be simulated from $U$ using $M$ without information loss).

- **More informative $U \succcurlyeq_I V$.** The embedding model $P_{U|X}$ is said to be "more informative" (or less noisy) than $P_{V|X}$ if and only if the embedding models always satisfy the inequality

$$I(Y;U) \geqslant I(Y;V), \quad \forall P_{YX} \in \mathcal{P}(\mathsf{Y} \times \mathsf{X}).$$

**Proposition 1** (Relationships of sufficiency and information). *The following relationships hold:*

(i) ***Sufficiency $\Rightarrow$ informativeness.*** *If the embedding model $P_{U|X}$ is sufficient for the embedding model $P_{V|X}$, i.e. $U \succcurlyeq_S V$, then $U \succcurlyeq_I V$. However, **Informativeness $\nRightarrow$ sufficiency.***

(ii) ***Informativeness $\Rightarrow$ higher capacity to distinguish concepts.*** *If the embedding model $P_{U|X}$ is more informative than embedding model $P_{V|X}$, i.e. $U \succcurlyeq_I V$, then*

$$KL\big(P_{U|Y}(\cdot|y_0)\|P_{U|Y}(\cdot|y_1)\big) \geqslant KL\big(P_{V|Y}(\cdot|y_0)\|P_{V|Y}(\cdot|y_1)\big),$$

*for any pair of concepts $(y_0, y_1) \in \mathsf{Y} \times \mathsf{Y}$ and all probability distributions $P_{YX}$.*

*Remark* 1. An immediate consequence of claim (i) is that the sufficient condition between embedding models implies that the embedding model $U$ is more informative than the embedding model $V$ relative to all target concepts in $\mathsf{Y}$ over all possible data distributions: $I(Y;U) \geqslant I(Y;V)$, for all probability distributions $P_{YX}$.

Although $U$ being more informative than $V$ does not necessarily imply $U \succcurlyeq_S V$ [57, 66]; (ii) states that being more informative ensures a higher statistical discrimination capacity between any pairs of target concepts (for further discussion, see Sec. B.2).

Motivated by the concepts of sufficiency and informativeness between embedding models, we can inquire about their statistical consequences for a learner conducting an inference task on these embeddings. More precisely, given a finite set of concepts $\mathsf{Y}$, **if $U \succcurlyeq_S V$, is the Bayes risk expected to be smaller when the inference is based on $U$ than when it is based on $V$?**

## 2.3 Comparing statistical experiments with embedding Models

The pursuit of comparing statistical experiments originated from the seminal paper by Bohnenblust, Shapley, and Sherman [16], followed by subsequent contributions by Blackwell [13, 14]. They formally established the relationships between sufficiency (Def. 1) and inference procedures.

In our framework, a statistical experiment [13] consists of a mathematical abstraction (see Appendix A for further details) intended to represent a downstream task where a learner aims at inferring a concept $y \in \mathsf{Y}$ from the embeddings $U$ or $V$. Deciding what embedder should be used to perform a given task is too general. In this work, we do not take into account the computational cost or the size of an embedder and solely focus on the following question:

> **What are the necessary and sufficient conditions that ensure that employing the embedding $U$ for any task $P_{YX}$ leads to lower risk compared to using the embedding $V$?**

Drawing parallels with the theoretical framework established for comparing statistical experiments, a relationship can be derived between the concept of sufficiency and the expected risk for a specific task (see Sec. B.5 for further discussion).

We concentrate on the scenario where $\mathsf{Y}$ consists of a finite number of concepts (e.g., classification tasks), as it is a significant case in its own right [104] and provide fundamental insights for the present work. The next Proposition states an important **relation between the concept of sufficiency and the expected Bayes risk on any classification task.**

**Proposition 2** (Comparison of embedding models through Bayes risks)**.** *Given two embedding models $P_{U|X} \in \mathcal{K}(\mathsf{U}|\mathsf{X})$ and $P_{V|X} \in \mathcal{K}(\mathsf{V}|\mathsf{X})$, the following statements are equivalent:*

*(i) The embedding model $P_{U|X}$ is sufficient relative to $P_{V|X}$, i.e. $U \succcurlyeq_S V$.*

*(ii) For all conditional probability measures $P_{Y|X}$ on finite alphabet $\mathsf{Y}$, the Bayes risks satisfy*

$$\inf_{\rho_U : \mathsf{U} \to \mathcal{P}(\mathsf{Y})} \Pr\left(\hat{Y}_U \neq Y\right) \leqslant \inf_{\rho_V : \mathsf{V} \to \mathcal{P}(\mathsf{Y})} \Pr\left(\hat{Y}_V \neq Y\right),$$

*where $\hat{Y}_U$ and $\hat{Y}_V$ are distributed according to $\rho_U(U)$ and $\rho_V(V)$, respectively.*

*Remark* 2. In other words, if we can fully simulate an embedder $V$ from another embedder $U$, the expected risk across all potential classification tasks cannot be greater when using $U$ compared to $V$. The proof of this Proposition is given in Sec. B.3. It is worth mentioning that various versions of this result are available in the literature [104]. However, our extension here, in a simpler setting, incorporates concepts and features into the experiment comparison framework.

## 2.4 Challenges in ranking embedding models and their deficiency

According to the notion of "sufficiency", we can distinguish the three following possibilities:

- Equivalence: $U \succcurlyeq_S V$ and $V \succcurlyeq_S U$ denoted $U \approx V$; $U$ and $V$ can simulate each other.
- Comparability: $U \succcurlyeq_S V$ but $V \not\succcurlyeq_S U$ only $V$ can be simulated from $U$.
- Non-comparability: $U \not\succcurlyeq_S V$ and $V \not\succcurlyeq_S U$, neither $U$ nor $V$ can simulate each other.

Our results up to now only account for the two first possibilities. However, two embedders are generally not comparable (Sec. B.4). This issue was addressed by Le Cam [63], who introduced the notion of "deficiency".

**Definition 2.** The deficiency $\delta(P_{U|X} \to P_{V|X})$ of $P_{V|X}$ relative to $P_{U|X}$ is defined as [63]

$$\delta(P_{U|X} \to P_{V|X}) \triangleq \inf_{M \in \mathcal{K}(\mathsf{V}|\mathsf{U})} \mathbb{E}\|M \circ P_{U|X} - P_{V|X}\|_{\mathrm{TV}},$$

where the infimum is taken over all Markov kernels (or transition probabilities) $M \in \mathcal{K}(\mathsf{V}|\mathsf{U})$, mapping stochastically $\mathsf{U}$ and $\mathsf{V}$, and $\delta$ measures error between the simulated and true embedders.

$\delta$ **indicates how well one model can be reconstructed from the other**, it induces a natural relaxation of the sufficiency where the reconstruction does not have to be perfect[2] for us to obtain guarantees on

---

[2]If $U \succcurlyeq_S V$, then $\delta(P_{V|X} \to P_{U|X}) = 0$, while if $U \not\succcurlyeq_S V$, then $\delta(P_{V|X} \to P_{U|X}) > 0$.

the downstream tasks performance (See Corollary 1). It avoids the non-comparability problem by evaluating **"how much information" we lose when passing from one model to the other one**.

Le Cam [63] showed that, for a given task $Y$, the deficiency $\delta(P_{U|Y} \to P_{V|Y})$ is directly related to the expected Bayes risks on the task (see Sec. B.6). We extend this result to the comparison of two embedding models $P_{U|X}$ and $P_{V|X}$ in a task-agnostic manner and build the relation to the expected Bayes risks for any classification task $Y$.

**Corollary 1.** *Given two embedding models $P_{U|X}$ and $P_{V|X}$ satisfying:*

*(i) The deficiency $\delta(P_{U|X} \to P_{V|X}) \leqslant \gamma$.*
*(ii) For any conditional distribution $P_{Y|X}$ on finite alphabets $\mathsf{Y}$,*

$$\inf_{\rho_U : \mathsf{U} \to \mathcal{P}(\mathsf{Y})} \Pr\left(\hat{Y}_U \neq Y\right) - \varepsilon \leqslant \inf_{\rho_V : \mathsf{V} \to \mathcal{P}(\mathsf{Y})} \Pr\left(\hat{Y}_V \neq Y\right).$$

*Statement (ii) implies (i) provided that $\gamma \geqslant 2|\mathsf{Y}|\varepsilon$ and conversely, (i) implies (ii) provided that $\gamma \leqslant \varepsilon$.*

The proof of this Corollary is relegated to Sec. B.3.

*Remark* 3. In particular, we can infer that for any classification task $Y$, the expected Bayes risk of the embedding model $U$, denoted by $\mathcal{R}_U$, is upper bounded by the expected Bayes risk of the embedding model $V$, denoted by $\mathcal{R}_V$:

$$\mathcal{R}_U - \mathcal{R}_V \leqslant \delta(P_{U|X} \to P_{V|X}), \quad \text{for all conditional distributions } P_{Y|X},$$

and similarly, $|\mathcal{R}_U - \mathcal{R}_V| \leqslant \max\left\{\delta(P_{U|X} \to P_{V|X}), \delta(P_{V|X} \to P_{U|X})\right\}$, for all conditional distributions $P_{Y|X}$. If both deficiencies are small, the resulting expected Bayes risks of the embedding models $U$ and $V$ will be close to each other for any target task $Y$.

## 3 Quantifying Information Sufficiency Between Embedding Models

We want to compare embedding models using the concept of deficiency, leveraging Prop. 2 and Corollary 1. These propositions suggest that the performance on any classification task of an embedding model $U$ relative to the model $V$ is bounded by $\delta(P_{U|X} \to P_{V|X})$. However, estimating the deficiency from data samples is notably challenging [95], and while upper bounds derivation exists, they do not necessarily make it tractable.

### 3.1 Estimating Information Sufficiency

The deficiency $\delta(P_{U|X} \to P_{V|X})$ between two embedding models $P_{U|X}$ and $P_{V|X}$, measures how well $U$ can be used to simulate $V$ using a Markov kernel $M \in \mathcal{K}(\mathsf{V}|\mathsf{U})$. This section aims to build a tractable proxy for this reconstruction cost. To this end, we estimate how much we can reduce the uncertainty about $Z$ by observing $U$ by learning an appropriate Markov kernel. This corresponds to the information sufficiency [29, 4] and can be interpreted as the information-theoretic counterpart of the deficiency. The information deficiency between $U$ and $V$ is then defined as:

**Definition 3** (Information sufficiency). The information sufficiency $\mathcal{I}_S(U \to V)$, relative to parametric classes of distributions $\mathcal{F}_\Theta(\mathsf{V})$ and $\mathcal{K}_\Theta(\mathsf{V}|\mathsf{U})$ (multivariate Gaussian mixtures [82]) is defined:

$$\mathcal{I}_S(U \to V) \triangleq \underbrace{\inf_{f \in \mathcal{F}_\Theta(\mathsf{V})} \mathbb{E}\left[-\log f(V)\right]}_{\text{Uncertainty of } V} - \underbrace{\mathbb{E}\left[\inf_{M \in \mathcal{K}_\Theta(\mathsf{V}|\mathsf{U})} \mathbb{E}\left[-\log M(V|U)|U\right]\right]}_{\text{Uncertainty when simulating } V \text{ from } U \text{ with } M}. \tag{1}$$

*Remark* 4. When the information sufficiency $\mathcal{I}_S(U \to V)$ is large, it signifies that $U$ offers a substantial amount of information to simulate $V$, a proxy for a small deficiency. Conversely, when $\mathcal{I}_S(U \to V)$ is lower, it implies that the channel $P_{V|Y}$ is subject to considerable noise or randomness, leading to a greater loss of statistical information.

We hence attempt to simulate $V$ from $U$ by learning a Markov kernel $M \in \mathcal{K}_\Theta(\mathsf{V}|\mathsf{U})$, via a mixture of multivariate Gaussians, and measure the uncertainty reduction it induces.

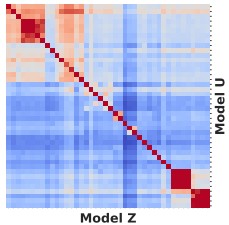

Figure 2: Pairwise $\mathcal{I}_S$ for text embedders.

**Pairwise embedder evaluation.** For set of embbeders $(Z_k)_k$ represented by their Markov kernels $\{P_{Z_k|X}\}_k$, we compute the pairwise information sufficiency $\mathcal{I}_S(Z_k \rightarrow Z_l)$. The pairwise information sufficiency matrix defines the adjacency matrix of a directed graph of embedders (Figure 2). Corollary 1 shows that embedders sharing high information sufficiency are expected to perform similarly on any downstream tasks, motivating the identification of communities in the graph. While the graph construction is in $\mathcal{O}(N^2)$; where $N$ is the number of embedders, it is in practice tractable for a reasonable number of embedders (refer to Sec. E.6) for more details).

**Practical embedding evaluation.** We construct the set of all information sufficiency using $Z_k$: $\mathcal{S}_{\mathcal{I}_S}(k) = \{\mathcal{I}_S(Z_k \rightarrow Z_l)\}_{l \neq k}$. We build our information sufficiency score ($\overline{\mathcal{I}_S}$ score) by taking the median of $\mathcal{S}_{\mathcal{I}_S}(k)$. Details on the $\overline{\mathcal{I}_S}$ score's estimation can be found in Sec. E.1.

## 4 Experimental Setup

We aim to evaluate the practical utility of the $\overline{\mathcal{I}_S}$ score to rank and select the best embedders for a given data distribution. We compare this ranking to those obtained on various downstream tasks. Our experimental protocol is divided into three main steps:

1. We evaluate the $\overline{\mathcal{I}_S}$ score of the models by identifying a large and diverse dataset that is supposed to be representative of the data distribution of interest.
2. We train a small feedforward neural network ($\rho_{Z_k}$) per embedder $P_{Z_k|X}$ to perform each downstream task and record its performances ($R^2$ score for regression, AUROC/accuracy for binary/multiclass classification).
3. We compare the models' performances on the downstream tasks and the $\overline{\mathcal{I}_S}$ score by measuring three types of correlations: the Pearson correlation, the Spearman correlation, and the Kendall-Tau coefficient.[3](See Sec. E.5 for additional baselines).

| | $\varrho_p$ | $\varrho_s$ | $\tau$ |
|---|---|---|---|
| Retrieval (15 datasets) | 0.89 | 0.89 | 0.69 |
| Classification (12 datasets) | 0.92 | 0.88 | 0.73 |
| Clustering (11 datasets) | 0.86 | 0.85 | 0.66 |
| STS (10 datasets) | 0.92 | 0.83 | 0.63 |
| Reranking (4 datasets) | 0.84 | 0.78 | 0.64 |
| Average (56 datasets) | **0.94** | **0.90** | **0.73** |
| Additional Classif (8 datasets) | 0.89 | 0.84 | 0.66 |

(a) NLP

| | $\varrho_p$ | $\varrho_s$ | $\tau$ |
|---|---|---|---|
| **A**bsorption (8 datasets) | - | 0.89 | 0.70 |
| **D**istribution (3 datasets) | - | 0.89 | 0.70 |
| **M**etabolism (8 datasets) | - | 0.94 | 0.79 |
| **E**xcretion (3 datasets) | - | 0.77 | 0.60 |
| **T**oxicity (9 datasets) | - | 0.92 | 0.75 |
| **ADMET** (31 datasets) | - | **0.94** | **0.80** |
| *DTI (1496 tasks) see Sec. D.4* | *-* | *0.88* | *0.70* |

(b) Molecular Modelling

(c) NLP

(d) Molecular Modeling

Figure 3: Correlation between $\overline{\mathcal{I}_S}$ scores and downstream task performances in (a) NLP and (b) Molecular Modelling. $\varrho_p$ is the Pearson correlation, $\varrho_s$ the spearman correlation, and $\tau$ is the Kendall-Tau coefficient. See Sec. C.3.1 for unaggregated results in NLP and Sec. D.3 in molecular modeling.

---

[3]For the experiments in molecular modeling, in each subset regression and classification tasks are mixed. Hence, we do not compute the Pearson correlation to avoid mixing scores obtained for different metrics.

# 5 Text Embeddings Evaluation

## 5.1 Experimental setting

**Embedders & Datasets.** We compared $34$ models with different training objectives, training datasets, and architectures. We included embedders derived from modern LLM such as LLaMA [106], Mistral [52], Gemma [102], Croissant [37] and T5 encoders [77]; common embedders derived from BERT architectures [31, 38, 85] or RobERTa [41] and embedders trained on specific embeddings objectives such Angle [64], Stella[4], E5 models [113], LaBSE [38]. A comprehensive list of the models can be found in Sec. C.1, Tab. 1 with their main characteristics and links to the Huggingface Hub for reproducibility. We used them to extract embeddings for many different datasets from the MTEB benchmark such as Banking77 [19], Sickr [122], Amazon polarity [72], SNLI [120] and IMDB [70]. We provide the datasets statistics in Sec. C.1, Tab. 2.

**Downstream tasks evaluation.** We rely on the results released on the MTEB leaderboard[5] and compare our rankings to the rankings and scores obtained by the different models on the different tasks. We evaluate additional tasks that are not included in the MTEB benchmark, such as tweet_eval [8, 74, 7, 109, 9], DAIR Emotion [92], agnews topic classification [123], Clinc intent detection [62] PAWS-X [118] and Rotten Tomatoes [79].

## 5.2 Model's Information Sufficiency analysis

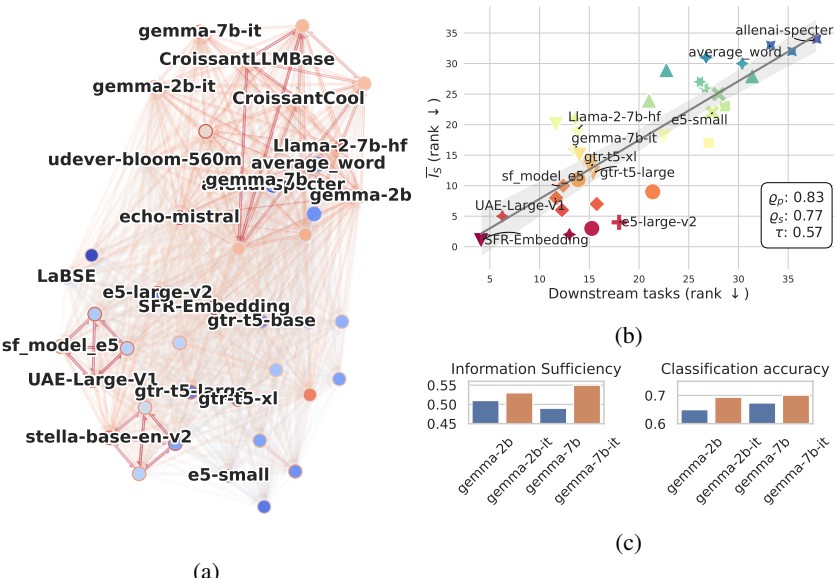

(a)

(b)

(c)

Figure 4: Figure 4a, presents the information sufficiency directed graph and the induced communities. Figure 4b displays the performance on additional downstream tasks and models not evaluated in the MTEB leaderboard. Figure 4c shows that instruction finetuning positively impacts the models' performance on the downstream tasks and that this improvement is captured by $\overline{\mathcal{I}_S}$.

**Correlation with downstream tasks performance.** The MTEB Benchmark offers a natural starting point to compare models' ranking according to their performance on downstream tasks and their $\overline{\mathcal{I}_S}$ score. In Figure 3c, we show that the $\overline{\mathcal{I}_S}$ score of an embedder correlates positively with its performance on a wide range of downstream tasks, from classification and similarity tasks to retrieval and clustering tasks. Overall, our $\overline{\mathcal{I}_S}$ score correlates strongly with MTEB's average score (Spearman correlation of $0.90$ and a Pearson correlation of $0.94$, see Figure 3c) and with the subtask

---

[4]https://huggingface.co/infgrad/stella-base-en-v2
[5]https://huggingface.co/spaces/mteb/leaderboard

performance Figure 3a). We extended our experiments to a more extensive set of models not included in the MTEB benchmark and observed a similar trend (Figure 4b). Per-datasets results are reported in Sec. C.3.1 and ablations in Sec. C.3.2. All our results show that our estimation of the information sufficiency between models is a good proxy for the performance of the models on a wide range of tasks.

**Embedder communities.** The pair-wise information sufficiency evaluation between the models can be used to cluster them into communities [15](Figure 4a, Figure 2)[6]. We observe that the extracted clusters group together models that are similar in their training objectives and architectures. LLM-based models such as LLaMA, Mistral, Gemma, and Croissant are clustered together, while BERT-based models share another cluster. Similarly, models trained specifically for embedding purposes, such as UAE-Large-V1 and ember-v1, are grouped together. This suggests that the ordering induced by information sufficiency is meaningful and can be used to identify models with similar properties and behaviors. Consistently with Corollary 1, we observe that the performance of the models on the downstream tasks is similar within the same cluster (Figure C.3.5). In addition, we found that it captures improvements by both steps of pretraining and instruction fine-tuning (Figure 4c, Sec. C.3.2)

# 6 Molecular Modeling

## 6.1 Experimental setting

**Embedders.** To process molecular data, embedders can leverage different representations of the molecules, providing an interesting benchmark to evaluate the $\overline{\mathcal{I}_S}$ score. We evaluated models derived from the molecular representation learning literature, summed up in Sec. D.1. We considered various input modalities such as string representations (SMILES [114], SELFIES [58]), 2D-graphs by using graph neural networks (GNNs), and 3D-representations (using the TorchMD-net architecture [80]). We added a randomly initialized baseline GNN model that was not trained on any dataset.

**Datasets.** To evaluate the information sufficiency between embedders, we compared the models on the ZINC 250k dataset[50], designed to gather compounds that could be relevant to a wide range of therapeutic projects. This dataset contains 250k commercially available compounds meant to be used in diverse therapeutic projects.

**Downstream tasks.** We evaluated the embedders on 31 downstream tasks extracted from the Therapeutic Data Commons [49] platform. This section focuses on ADMET tasks (Absorption, Distribution, Metabolism, Excretion, and Toxicity). Results on Drug-Target interaction tasks can be found in Sec. D.4. Datasets collected are split into a training, validation, and test set, following the scaffold-split strategy, further described in see Sec. D.3.

## 6.2 Model's Information Sufficiency analysis

**Global results.** The $\overline{\mathcal{I}_S}$ score ranking is consistent with the results of the embedders on the ADMET downstream tasks, achieving a Spearman correlation of 0.95 and a Kendall-tau coefficient of 0.80, as reported in Figure 3d. Detailed results for each of the 31 tasks are available in Sec. D.3 in Tab. 6. Table 3b shows the correlation between the $\overline{\mathcal{I}_S}$ score rankings and the performances obtained on the ADMET tasks within each category. High correlations are achieved within most task categories, especially when large tasks are available (containing an important number of molecules). On excretion tasks, the correlation is lower (below 0.8), which can be explained by the fact that these tasks are the most challenging regression tasks available, where the fine-tuned models reach the lowest $R^2$ scores between 0 and 0.2 (see Sec. D.3).

**Most / Least promising models.** We observe in Figure 5b that the most promising models are the (Chem)Bert-MTR models[3][7] and MolR[112], the former trained on SMILES representations to predict a variety of computationally available molecular properties, and the latter trained on 2D graphs to preserve equivalence of molecules w.r.t chemical reactions. Surprisingly, these models share high predictive mutual information (being assigned to the same Louvain community in Figure 5a),

---

[6]We rely on the Louvain community detection implementation from networkx[43]

[7]BertMTR-$X$M stands for a (Chem)Bert-MTR model trained on $X$M molecules.

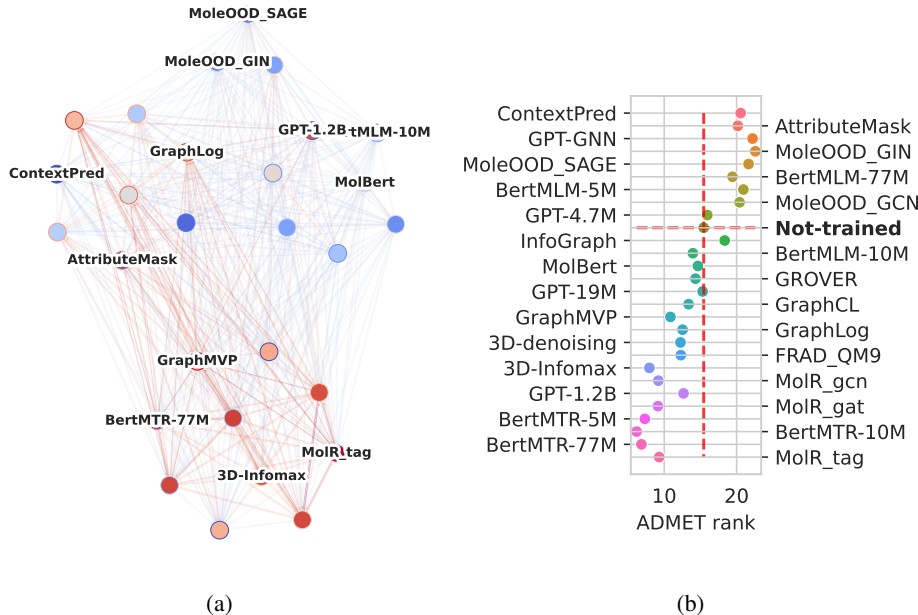

(a)                                     (b)

Figure 5: (a) Pairwise information sufficiency graph between the embedders. The center color represents the ability to simulate other models, while the surrounding colors represent the ability to be simulated by other models. Red indicates a high ability to simulate or be simulated, while blue indicates a low ability. (b) Mean rank of the models (ordered by $\overline{\mathcal{I}_S}$ score) on downstream tasks.

suggesting that they capture similar information despite significant differences in their training methods. These models also appear to be the most competitive on the ADMET tasks. On the other hand, and consistently with Sun *et al.* [99]'s observation, training methods for 2D-GNNs such as following attribute masking and context prediction objective are deemed as the least informative according to the $\overline{\mathcal{I}_S}$ score. This is explained by the simplicity of these pretraining objectives for this data modality. These methods are also among the least competitive methods on the ADMET downstream tasks.

**NLP-inspired models.** (*Chem*)Bert-MLM [3], MolBert [36] and (*Chem*)GPT[40] leverage masked language model objective applied to string representations (SMILES and SELFIES). Unsurprisingly, as seen in Figure 5a, these models are clustered, suggesting they capture similar information. However, they fail to simulate other models in the pool, resulting in low $\overline{\mathcal{I}_S}$ scores, a result consistent with the known limitations of these pretraining objectives [23, 105]. A noticeable exception is (*Chem*)GPT-1.2B (the biggest model of the pool by far), which displays a significantly higher $\overline{\mathcal{I}_S}$ score.

**"Not-trained" GNN.** Figure 5b helps visualize the performances of the different models relative to our baseline "Not-trained" GNN. Surprisingly, some models are ranked less promising than this baseline by the $\overline{\mathcal{I}_S}$ score. However, all of these less promising models obtain poorer performances on the downstream tasks. Similarly, except for InfoGraph [98], every model ranked more promising than the "Not-trained" GNN baseline and obtained better results on ADMET tasks. This surprising result validates evaluation of the $\overline{\mathcal{I}_S}$ score w.r.t this baseline.

## 7   Limitations and Conclusions

We proposed a principled approach to embedding model evaluation by framing model ranking as a variation of comparing statistical experiments. Utilizing concepts of sufficiency, informativeness, and deficiency, we developed mathematically grounded metrics for pairwise comparisons between embedders without relying on labeled data in downstream tasks. Our tractable relaxation, termed information sufficiency, demonstrated strong correlations with rankings based on downstream task performance in extensive experiments. Although successful, our method still has at least two primary

limitations. First, its effectiveness depends on the number and diversity of available embedders (see Sec. E.4). Future work could explore using randomly initialized embedders (random projections) instead of pre-trained ones. Second, we can enhance our proxy for predicting the deficiency between models by exploring better methods (e.g., estimating the $f$-divergence) to directly learn the Markov kernel that minimizes the total variation distance, which we leave for future research.

## Acknowledgments

This work was granted access to the HPC resources of IDRIS under the allocation 2023-AD011013290R2 made by GENCI, and enabled by support provided by Calcul Quebec and the Digital Research Alliance of Canada. We warmly thank Heitor Rapela, Banafsheh Karimian, and Eric Aubinais for their advice and comments about our work. We also owe a special highlight to Loïc Fosse for the many discussions and hindsights he provided and for the subsequent follow-up projects.

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

# Appendix

## Table of Contents

## A    Background and Notation

We consider all alphabets to be standard Borel [28] (i.e., isomorphic to a Borel subspace of a Polish space), encompassing virtually all practical scenarios. Each such space $\mathsf{Y}$ is equipped with its Borel $\sigma$-algebra $\mathcal{B}(\mathsf{Y})$. The set of all probability measures on $\mathsf{Y}$ is denoted by $\mathcal{P}(\mathsf{Y})$. The total variation distance between $P$ and $Q$ in $\mathcal{P}(\mathsf{Y})$ is defined as

$$\|P - Q\|_{\mathrm{TV}} = \sup_{\mathcal{A} \in \mathcal{B}(\mathsf{Y})} |P(\mathcal{A}) - Q(\mathcal{A})|, \tag{2}$$

and the Kullback–Leibler divergence is defined by

$$\mathrm{KL}(P\|Q) = \begin{cases} \int_{\mathsf{X}} \log \frac{\mathrm{d}P}{\mathrm{d}Q} \mathrm{d}P & \text{if } P \ll Q \\ +\infty & \text{otherwise.} \end{cases} \tag{3}$$

Given a joint probability measure $P_{XY}$ in $\mathcal{P}(\mathsf{X} \times \mathsf{Y})$ induced by two random variables $X \in \mathsf{X}$ and $Y \in \mathsf{Y}$ with product measures $P_X P_Y$, the Mutual Information is defined as $I(X;Y) = \mathrm{KL}(P_{XY}\|P_X P_Y)$. If $P_X \in \mathcal{P}(\mathsf{X})$ is a probability measure induced by $X \in \mathsf{X}$, the Differential Entropy is defined by

$$h(X) = -\int_{\mathsf{X}} \log \frac{\mathrm{d}P_X}{\mathrm{d}\mu} \, dP_X, \tag{4}$$

where $\mu$ denotes the Lebesgue measure. Similarly, it is possible to define the conditional entropy of $Y$ given $X$ which is denoted by $h(Y|X)$. The mutual information satisfies the identities $I(X;Y) = h(Y) - h(Y|X) = h(X) - h(X|Y)$ and $h(Y|X) \leqslant h(Y)$ (see [27] for further details).

A Markov (or transition probability) kernel between $\mathsf{X}$ and $\mathsf{Y}$ is a mapping $T : \mathcal{B}(\mathsf{Y}) \times \mathsf{X} \to [0,1]$, satisfying $T(\cdot|x) \in \mathcal{P}(\mathsf{Y})$ for all $x \in \mathsf{X}$ and $T(\mathcal{B}|\cdot)$ being a measurable function on $\mathsf{X}$ for any $B \in \mathcal{B}(\mathsf{Y})$. The space of all such $T$ is denoted by $\mathcal{K}(\mathsf{Y}|\mathsf{X})$. In cases where both $\mathsf{Y}$ and $\mathsf{X}$ are finite, any $\mathcal{K}(\mathsf{Y}|\mathsf{X})$ is represented as a stochastic matrix with elements $T(y|x)$, $(x,y) \in \mathsf{X} \times \mathsf{Y}$. Every $T \in \mathcal{K}(\mathsf{Y}|\mathsf{X})$ induces a mapping $\mathcal{P}(\mathsf{X}) \longrightarrow \mathcal{P}(\mathsf{Y})$, denoted by $T$, mapping any $P \in \mathcal{P}(\mathsf{X})$ to $Q = T \circ P \in \mathcal{P}(\mathsf{Y})$, where

$$Q(B) = (T \circ P)(B) \triangleq \int_{\mathsf{X}} T(B|x) P(\mathrm{d}x), \quad \forall B \in \mathcal{B}(\mathsf{Y}). \tag{5}$$

We denote the composition of Markov kernels by juxtaposition: for $M \in \mathcal{K}(\mathsf{Z}|\mathsf{Y})$ and $T \in \mathcal{K}(\mathsf{Y}|\mathsf{X})$, their composition $M \circ T \in \mathcal{T}(\mathsf{Z}|\mathsf{X})$ is defined by

$$(M \circ T)(Z|x) \triangleq \int_{\mathsf{Y}} M(Z|y) T(\mathrm{d}y|x), \quad \forall x \in \mathsf{X}, Z \in \mathcal{B}(\mathsf{Z}). \tag{6}$$

We define the average of the total variation distance between two Markov kernels $T, T' \in \mathcal{K}(\mathsf{Y}|\mathsf{X})$ as follows:

$$\mathbb{E}\|T - T'\|_{\mathrm{TV}} \triangleq \mathbb{E}\|T(\cdot|X) - T'(\cdot|X)\|_{\mathrm{TV}}. \tag{7}$$

A statistical model is a triple $\mathcal{M}_U \equiv (\mathsf{U}, \mathcal{B}(\mathsf{U}), (P_{U|Y}(\cdot|y) : y \in \mathsf{Y}))$, where $(\mathsf{U}, \mathcal{B}(\mathsf{U}))$ is a sample space; $\mathsf{Y}$ is a concept space, and $P_{U|Y} : \mathcal{B}(\mathsf{U}) \times \mathsf{Y} \to [0,1]$ is a Markov kernel (or transition probability).

## B    Proofs Theoretical Results

**Proposition** (Relationships of sufficiency and information)**.** *The following relationships hold:*

   (i) ***Sufficiency*** $\Rightarrow$ ***informativeness.*** *If the embedding model $P_{U|X}$ is sufficient for the embedding model $P_{V|X}$, i.e. $U \succcurlyeq_S V$, then $U \succcurlyeq_I V$. However, **Informativeness** $\nRightarrow$ **sufficiency.***

   (ii) ***Informativeness*** $\Rightarrow$ ***higher capacity to distinguish concepts.*** *If the embedding model $P_{U|X}$ is more informative than embedding model $P_{Z|X}$, i.e. $U \succcurlyeq_I V$, then*

$$KL(P_{U|Y}(\cdot|y_0)\|P_{U|Y}(\cdot|y_1)) \geqslant KL(P_{V|Y}(\cdot|y_0)\|P_{V|Y}(\cdot|y_1)),$$

   *for any pair of concepts $(y_0, y_1) \in \mathsf{Y} \times \mathsf{Y}$ and all probability distributions $P_{YX}$.*

## B.1 Proof Proposition 1

*Proof.* It is immediate to check that the data-processing inequality and the Markov chain $Y \leftrightarrow X \leftrightarrow U \leftrightarrow V$ implies the relation in claim (i). On the other hand, the non-equivalence is proved by means of an explicit counterexample [57]. Given any $0 < p < 1/2$ with $\bar{p} = 1 - p$. For some $\epsilon, \delta > 0$, consider two discrete embedding models defined by the following matrices:

$$P_{V|X} = 1/2 \begin{pmatrix} 1+\epsilon & 1-\epsilon \\ 1 & 1 \end{pmatrix} P_{U|X} \quad \text{with} \quad P_{U|X} = \begin{pmatrix} p & \bar{p} \\ p+\delta & \bar{p}-\delta \end{pmatrix}. \tag{8}$$

By taking $\epsilon, \delta > 0$ small enough, both $P_{U|X}$ and $P_{V|X}$ are stochastic matrices. It follows that $P_{V|X}$ is not a degraded version of $P_{U|X}$ but provided that $\epsilon, \delta$ are sufficient small, the embedding model $P_{U|X}$ is more informative than $P_{V|X}$, which proves the claim.

In order to show (ii), let $P_{U|Y}$ and $P_{V|Y}$ be the corresponding probability measures induced by $P_{X|U}(\cdot|u)$ via the embedding models:

$$P_{U|Y}(U|y) = \int_X P_{U|X}(U|x) P_{X|Y}(\mathrm{d}x|y), \quad \forall y \in \mathsf{Y}, \, U \in \mathcal{B}(\mathsf{U}), \tag{9}$$

and

$$P_{V|Y}(V|y) = \int_X P_{V|X}(V|x) P_{X|Y}(\mathrm{d}x|y), \quad \forall y \in \mathsf{Y}, \, V \in \mathcal{B}(\mathsf{V}), \tag{10}$$

for any $y \in \mathsf{Y} = \{y_0, y_1\}$. For a $0 \leqslant \lambda \leqslant 1$, let $P_{X|Y}(\cdot|y_0) \in \mathcal{P}(\mathsf{X})$ and $P_{X|Y}(\cdot|y_1) \in \mathcal{P}(\mathsf{X})$ be two arbitrary probability measures on $\mathsf{X}$. Let $P_{X|Y}(X|y)$ be defined by

$$P_{X|Y}(X|y) = \mathbb{1}[y = y_0] P_{X|Y}(X|y_0) + \mathbb{1}[y = y_1] P_{X|Y}(X|y_1).$$

By replacing it into equations (9) and (10), we obtain

$$P_{U|Y}(U|y) = \mathbb{1}[y = y_0] P_{U|Y}(U|y_0) + \mathbb{1}[y = y_1] P_{U|Y}(U|y_1) \tag{11}$$
$$P_{V|Y}(V|y) = \mathbb{1}[y = y_0] P_{V|Y}(V|y_0) + \mathbb{1}[y = y_1] P_{V|Y}(V|y_1) \tag{12}$$

and let $P_Y(y_0) = \lambda$ and $P_Y(y_1) = 1 - \lambda$. The above probability measures correspond to a quadruple of random variables: $(Y_\lambda, X_\lambda, U_\lambda, V_\lambda) \in \mathcal{P}(\mathsf{Y} \times \mathsf{X} \times \mathsf{U} \times \mathsf{V})$. Consider the function $f(\lambda)$ defined by

$$f(\lambda) = I(Y_\lambda; U_\lambda) - I(Y_\lambda; V_\lambda).$$

It is not difficult to check that $f(\lambda) \geqslant 0$ for all $0 \leqslant \lambda \leqslant 1$, and $f(0) = 0$ which requires that $f'(0) \geqslant 0$. By taking the differentiation, we obtain

$$f'(0) = \mathrm{KL}\big(P_{U|Y}(\cdot|y_0)\|P_{U|Y}(\cdot|y_1)\big) - \mathrm{KL}\big(P_{V|Y}(\cdot|y_0)\|P_{V|Y}(\cdot|y_1)\big) \geqslant 0,$$

which implies the claim (ii). This concludes the proof. $\qquad\square$

## B.2 Comments about capacity to distinguish concepts

Notice that the KL divergence between the induced distributions of the resulting embedding is not less for the embedding model $U$ than $Z$. Indeed, consider the case of binary classification $\mathsf{Y} = \{y_0, y_1\}$ with uniformly distributed concepts. Pinsker's inequality [108] together with claim (iii) imply

$$\mathrm{KL}\big(P_{U|Y}(\cdot|y_0)\|P_{U|Y}(\cdot|y_1)\big) \geqslant 2\|P_{V|Y}(\cdot|y_0) - P_{V|Y}(\cdot|y_1)\|_{\mathrm{TV}}^2.$$

From which, it is easy to verify that the accuracy of the expected Bayes accuracy of the optimal classifier based on $V$ is upper bounded by [108, Lemma 2.1]:

$$\sup_\psi \Pr(\psi(V) = Y) \leqslant 1 - \frac{1}{2} \exp\big(-\mathrm{KL}\big(P_{U|Y}(\cdot|y_0)\|P_{U|Y}(\cdot|y_1)\big)\big),$$

where the exponent in the upper bound is subject to the discriminating capacity through the KL divergence of the embedding model $U$ on $\mathsf{U}$.

### B.3 Proof of Proposition 2 and Corollary 1

We begin with the proof of Proposition 2.

*Proof.* Clearly, the assumption (i) implies the statement (ii) by Data-Processing. Conversely, let us assume point (ii) holds. This means that, for every probbaility distribution $P_X$ and all conditional probability distributions $P_{Y|X}$, there exists $\rho_U : \mathsf{U} \to \mathcal{P}(\mathsf{Y})$ such that

$$\sum_{(y,x)\in\mathsf{Y}\times\mathsf{X}} P_{YX}(y,x) \int_{\mathsf{U}} \rho_U(y|u) P_{U|X}(\mathrm{d}u|x) \geqslant$$

$$\sup_{\rho_V} \sum_{(y,x)\in\mathsf{Y}\times\mathsf{X}} P_{YX}(y,x) \int_{\mathsf{V}} \rho_V(y|v) P_{V|X}(\mathrm{d}v|x), \tag{13}$$

where $\rho_V : \mathsf{V} \to \mathcal{P}(\mathsf{Y})$ is a (possibly randomized) inference procedure is transition probabilities, which the learner can optimize to maximize the guessing probability.

Let the decision rule $\rho_V(y|v) = \mathbb{1}[v \in A_y]$ for any partition $\{A_y\}_{y\in\mathsf{Y}}$ of $\mathsf{V}$ with $A_y \in \mathcal{B}(\mathsf{V})$. Then, for any $P_{Y|X}$, expression (13) implies the existence there exists $\rho_U(y|u)$ such that

$$\sum_{(y,x)\in\mathsf{Y}\in\mathsf{X}} P_{YX}(y,x) \left[ \int_{\mathsf{V}} P_{V|X}(\mathrm{d}v|x)\mathbb{1}[v \in A_y] - \int_{\mathsf{U}} \rho_U(y|u)P_{U|X}(\mathrm{d}u|x) \right] \tag{14}$$

$$= \sum_{(y,x)\in\mathsf{Y}\times\mathsf{X}} P_{YX}(y,x) \left[ \int_{A_y} P_{V|X}(\mathrm{d}v|x) - \int_{\mathsf{U}} \rho_U(y|u)P_{U|X}(\mathrm{d}u|x) \right] \leqslant 0. \tag{15}$$

However, we can rewrite the last expression as:

$$\sup_{P_{Y|X}} \inf_{\rho_U} \sum_{(y,x)\in\mathsf{Y}\times\mathsf{X}} P_{YX}(y,x) \left[ \int_{A_y} P_{V|X}(\mathrm{d}v|x) - \int_{\mathsf{U}} \rho_U(y|u)P_{U|X}(\mathrm{d}u|x) \right] \leqslant 0. \tag{16}$$

By applying the minimax theorem [87], it is possible to exchange the order of the inf and the sup, which yields:

$$\inf_{\rho_U} \sup_{\{A_y\}} \mathbb{E}\left[ \sup_{P_{Y|X}} \sum_{y\in\mathsf{Y}} P_{Y|X}(y|X)\Gamma\big((y,X),\rho_U\big) \right] \leqslant 0 \tag{17}$$

$$\Gamma\big((y,x),\rho_U\big) \triangleq \left[ \int_{A_y} P_{V|X}(\mathrm{d}v|x) - \int_{\mathsf{U}} \rho_U(y|u)P_{U|X}(\mathrm{d}u|x) \right]. \tag{18}$$

We observe that

$$\sum_{y\in\mathsf{Y}} \Gamma\big((y,x),\rho_U\big) = 0, \tag{19}$$

for each $x \in \mathsf{X}$ and thus,

$$\max_{y\in\mathsf{Y}} \Gamma\big((y,x),\rho_U\big) \geqslant 0, \tag{20}$$

where the equality holds if and only if $\Gamma\big((y,x),\rho_U\big) = 0$ for all $y \in \mathsf{Y}$, for each $x \in \mathsf{X}$, since by contradiction otherwise

$$\sum_{y\in\mathsf{Y}} \Gamma\big((y,x),\rho_U\big) < 0. \tag{21}$$

Therefore,

$$\inf_{\rho_U} \sup_{\{A_y\}} \mathbb{E}\left[ \sup_{P_{Y|X}} \sum_{y\in\mathsf{Y}} P_{Y|X}(y|X)\left( \int_{A_y} P_{V|X}(\mathrm{d}v|X) - \int_{\mathsf{U}} \rho_U(y|u)P_{U|X}(\mathrm{d}u|X) \right) \right]$$

$$= \inf_{\rho_U} \sup_{\{A_y\}} \mathbb{E}\left[ \max_{y\in\mathsf{Y}} \Gamma\big((y,X),\rho_U\big) \right], \tag{22}$$

which means the maximum is achieved by degenerate random variables $Y = f(X)$ achieving the maximum for each $x \in \mathsf{X}$. Consequently, we have that

$$\inf_{\rho_U} \mathbb{E}\|P_{V|X} - \rho_U \circ P_{U|X}\|_{\text{TV}} \quad = \quad \inf_{\rho_U} \sup_{\{A_y\}} \mathbb{E}\left[\max_{y \in \mathsf{Y}} \Gamma\big((y, X), \rho_U\big)\right] = 0, \tag{23}$$

hence $\inf_{\rho_U} \mathbb{E}\|P_{V|X} - \rho_U \circ P_{U|X}\|_{\text{TV}} = 0$, and so the existence of the transition probability $\rho_U$ such that $P_{U|X}$ is sufficient for $P_{V|X}$. This concludes the proof of the Proposition 2. $\qquad \square$

We now show the proof of Corollary 1.

*Proof.* Continuing from the proof of Proposition 2, which remains unchanged, until one demonstrates that for $\varepsilon > 0$,

$$\inf_{\rho_U} \sup_{\{A_y\}} \mathbb{E}\left[\max_{y \in \mathsf{Y}} \Gamma\big((y, X), \rho_U\big)\right] \leqslant \varepsilon. \tag{24}$$

To proceed from this point, let us now examine the following quantity:

$$\sum_{y \in \mathsf{Y}} |\Gamma\big((y, x), \rho_U\big)|.$$

The above quantity is the induced $\ell_1$-norm distance between $\int_{\mathcal{A}_y} P_{V|X}(\mathrm{d}v|x)$ and $\int_{\mathsf{U}} \rho_U(y|u) P_{U|X}(\mathrm{d}u|x)$. Since, for all $x \in \mathsf{X}$,

$$\sum_{y \in \mathsf{Y}} \Gamma\big((y, x), \rho_U\big) = 0,$$

we have that

$$\sum_{y \in \mathsf{Y}} |\Gamma\big((y, x), \rho_U\big)| = 2 \sum_{y \in \mathsf{Y}: \Gamma((y,x), \rho_U) \geqslant 0} \Gamma\big((y, x), \rho_U\big), \quad \text{for all } x \in \mathsf{X}$$

which implies that, for the strategy $\rho_U$ achieving the left-hand side of Eq. (24),

$$\sup_{\{A_y\}} \mathbb{E} \sum_{y \in \mathsf{Y}} |\Gamma\big((y, X), \rho_U\big)| \leqslant 2|\mathsf{Y}| \sup_{\{A_y\}} \mathbb{E}\left[\max_{y \in \mathsf{Y}} \Gamma\big((y, X), \rho_U\big)\right] \leqslant 2|\mathsf{Y}|\varepsilon, \quad \text{for all } x \in \mathsf{X}$$

Hence,

$$\sup_{\{A_y\}} \mathbb{E} \sum_{y \in \mathsf{Y}} \left|\int_{\mathcal{A}_y} P_{V|X}(\mathrm{d}v|X) - \int_{\mathsf{U}} \rho_U(y|u) P_{U|X}(\mathrm{d}u|X)\right| \leqslant 2|\mathsf{Y}|\varepsilon.$$

This concludes the proof of the Corollary 1. $\qquad \square$

### B.4 Example of comparisons of statistical experiments

**Example 1** (Statistical experiments with Gaussian embedding models). *Let $U|x$ and $V|x$ be independently normally distributed as $\mathcal{N}(x, \sigma^2)$ and $\mathcal{N}(x, \epsilon^2 \sigma^2)$, respectively, with $0 < \epsilon < 1$.*

- *Case of $\sigma^2 = \sigma_0^2$ known. Here $U \succcurlyeq_S V$ since $V + \nu|x$ has the same distribution as $U|x$ when $\nu \sim \mathcal{N}(0, (1 - \epsilon^2)\sigma_0^2)$. That is $V$ is strictly more informative than $U$. However, $U$ is strictly more informative than $V$.*

- *Case of $x = 0$ known. One can observe that $U \approx V$ since the variables $V/\epsilon$ have the same distribution as the $U$, and the variables $\epsilon U$ have the same distribution as the $V$.*

- *Case of $x$ and $\sigma$ unknown. Surprisingly, in this case $U$ and $V$ are not comparable.*

## B.5 Sufficiency and Inference Procedures with Embedding Models [13, 63]

**Proposition 3** (Sufficiency and risks of a given task on embedding models [13, 63]). *An embedding model $P_{U|Y} \in \mathcal{K}(\mathsf{U}|\mathsf{Y})$ is deemed to be sufficient for another one $P_{V|Y} \in \mathcal{K}(\mathsf{V}|\mathsf{Y})$ if and only if, for any bounded loss function $\ell$ where $\|\ell\|_\infty \leqslant 1$, and for any inference procedure $\rho_V : \mathsf{V} \to \mathsf{Y}$, there exists a inference procedure (possibly randomized) $\rho_U : \mathsf{U} \to \mathcal{P}(\mathsf{Y})$ such that the resulting statistical risks satisfy*

$$\mathcal{R}_y(P_{U|Y}, \rho_U, \ell) \leqslant \mathcal{R}_y(P_{V|Y}, \rho_V, \ell), \quad \text{for all } y \in \mathsf{Y}. \tag{25}$$

*Here we denote by $\mathcal{R}_y(P_{U|Y}, \rho_Y, \ell)$ and $\mathcal{R}_y(P_{V|Y}, \rho_V, \ell)$ the statistical risks for the corresponding inference frameworks, respectively.*

*Remark* 5. The restriction $\|\ell\|_\infty \leqslant 1$ is irrelevant here. However, we opt for simplicity and limit our focus to situations where one encounters dominated statistical models with Polish sample spaces. In essence, various extensions do not significantly alter the conceptual aspects of the underlying statistical problem (see [103] for further details). Rather, they primarily reflect the complexity of its measure-theoretic formulation.

## B.6 Deficiency and Expected Risk [63]

In 1964, Le Cam [63] clarified the relationship between the sufficiency of an embedding model on a given task and its expected risk on this task. The following theorem provides a formal statement of this relationship.

**Theorem 1** (Le Cam [63]). *Let $\varepsilon > 0$ be fixed. Then, $\delta(P_{U|Y} \to P_{V|Y}) < \varepsilon$ if and only if, for any bounded loss function $\ell$ where $\|\ell\|_\infty \leqslant 1$, and for any inference procedure $\rho_V$ using the embedding model $P_{V|Y}$, there exists a inference procedure (possibly randomized) $\rho_U$ based the embedding model $P_{U|Y}$ such that the risks satisfy $\mathcal{R}_y(P_{U|Y}, \rho_U, \ell) - \varepsilon \leqslant \mathcal{R}_y(P_{V|Y}, \rho_V, \ell), \quad \text{for all } y \in \mathsf{Y}.$*

# C NLP Experiment Details

In this section, we provide all the necessary experimental details to reproduce the experiments in NLP. For the $\overline{\mathcal{I}_S}$ score estimation, please see Appendix E. First, we detail the models and datasets used in the experiments. We provide the training details of the downstream tasks, and finally, we present the comprehensive results of the NLP experiments.

## C.1 Models and Datasets statistics

In Tab. 1, we provide the metadata of the models used in the NLP experiments and their scores on the MTEB benchmark when they exist. We provide in Tab. 2 the statistics of the datasets used to evaluate the $\overline{\mathcal{I}_S}$ score.

## C.2 Downstream tasks training details.

All the downstream tasks are trained in the exact same way. We use a dense classifier with two hidden layers of dimension 256 and train for two epochs using ADAM [56] with a learning rate of $10^{-3}$, on the official training set and evaluated on either the validation or test set when they are available (with respect to the Huggingface datasets). We do not perform early stopping or selection using the validation set.

## C.3 NLP Comprehensive Results

We provide in this section the unaggregated results for the main NLP experiments presented in the main text, then we provide numerous ablation studies and additional results to address different aspects of the $\overline{\mathcal{I}_S}$ score of the embeddings in NLP.

### C.3.1 Full MTEB Benchmark Results

The strength of the MTEB benchmark is that it evaluates embedders on a very large and diverse set of downstream tasks. We provide an Tab. 4 and Figure 6 the full results of the MTEB benchmark (English) for the models used in the NLP experiments.

Table 1: Metadata of the evaluated models and their information sufficiency.

| Model | Dim. | Max Tokens | $\bar{I}_S$ |
|---|---|---|---|
| SFR-Embedding-Mistral | 4096 | 32768 | **0.59** |
| echo-mistral-7b-instruct-lasttoken | 4096 | 32768 | **0.58** |
| stella-base-en-v2 | 768 | 512 | **0.57** |
| e5-large-v2 | 1024 | 512 | **0.57** |
| GritLM-7B | 4096 | 32768 | **0.56** |
| ember-v1 | 1024 | 512 | **0.56** |
| gte-large | 1024 | 512 | **0.56** |
| UAE-Large-V1 | 1024 | 512 | **0.55** |
| gte-base | 768 | 512 | **0.55** |
| sf_model_e5 | 1024 | 512 | **0.55** |
| GIST-Embedding-v0 | 768 | 512 | **0.55** |
| gtr-t5-large | 768 | 512 | **0.55** |
| gtr-t5-xl | 768 | 512 | **0.55** |
| bge-base-en-v1.5 | 768 | 512 | **0.54** |
| sentence-t5-large | 768 | 512 | **0.54** |
| gemma-7b-it | N/A | N/A | **0.54** |
| gte-tiny | 384 | 512 | **0.53** |
| gtr-t5-base | 768 | 512 | **0.53** |
| Llama-2-7b-hf | N/A | N/A | **0.53** |
| sentence-t5-xl | 768 | 512 | **0.53** |
| gemma-2b-it | N/A | N/A | **0.52** |
| e5-small | 384 | 512 | **0.52** |
| bge-micro-v2 | 384 | 512 | **0.51** |
| all-distilroberta-v1 | N/A | N/A | **0.51** |
| multilingual-e5-small | 384 | 512 | **0.51** |
| msmarco-bert-co-condensor | 768 | 512 | **0.51** |
| sup-simcse-bert-base-uncased | 768 | 512 | **0.50** |
| all-MiniLM-L6-v2 | 384 | 512 | **0.50** |
| all-mpnet-base-v2 | 768 | 514 | **0.49** |
| udever-bloom-560m | 1024 | 2048 | **0.49** |
| LaBSE | 768 | 512 | **0.47** |
| average_word_embeddings_komninos | 300 | N/A | **0.42** |
| average_word_embeddings_glove.6B.300d | 300 | N/A | **0.41** |
| allenai-specter | 768 | 512 | **0.38** |

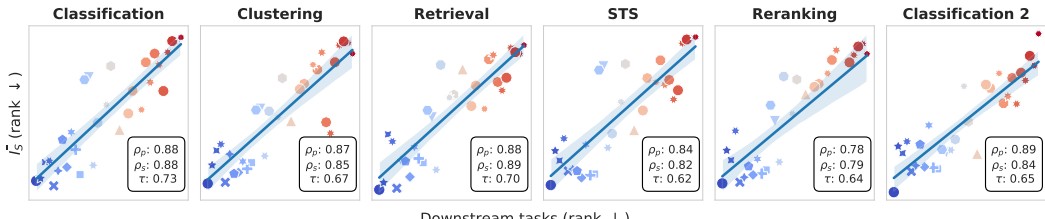

Figure 6: Correlations between rankings on different subtasks and their $\overline{\mathcal{I}_S}$ score ranking.

Table 2: Statistics of the datasets used as umbrella datasets for $\overline{\mathcal{I}_S}$ informativeness evaluation.

| Dataset | Split | Size |
|---|---|---|
| ag_news | test | 7600 |
| | train | 120000 |
| amazon_polarity | test | 100000 |
| banking77 | test | 3080 |
| biosses-sts | test | 182 |
| paws-x;en | test | 2000 |
| | train | 49401 |
| | validation | 2000 |
| rotten_tomatoes | test | 1066 |
| | train | 8530 |
| | validation | 1066 |
| sickr-sts | test | 6077 |
| snli | test | 13132 |
| | validation | 13134 |
| sst2 | test | 1821 |
| | train | 67349 |
| | validation | 872 |
| sts12-sts | test | 4946 |
| sts13-sts | test | 2638 |
| sts14-sts | test | 6351 |
| sts15-sts | test | 5170 |
| stsbenchmark-sts | test | 2552 |
| | validation | 2910 |
| tweet_eval;emoji | test | 50000 |
| | train | 45000 |
| | validation | 5000 |
| tweet_eval;emotion | test | 1421 |
| | train | 3257 |
| | validation | 374 |
| tweet_eval;sentiment | test | 12284 |
| | train | 45615 |
| | validation | 2000 |
| wiki-paragraphs | validation | 100000 |

We obtain significant positive correlations in all categories of downstream tasks. We noticed that we obtained significantly poorer results on STS, Clustering, and Reranking tasks than on classification tasks. We believe this behavior is due to the nature of these tasks. Indeed, they do not rely on training an additional model on top of the embeddings but rather directly use the embeddings as is in dot products or similarity measures. An embedder could produce very informative embeddings, *i.e.*, it is possible to extract the useful information using a small model, and at the same time, these embeddings not be adequate for dot-product-based similarity measures. We believe further investigation is needed to understand the behavior of the models on these tasks. Especially to see if training a small model on top of the embeddings can improve the performance of these tasks.

### C.3.2 Ablation studies

Many factors can impact the estimation of the $\overline{\mathcal{I}_S}$ score of the models, such as the dimensions of the different embeddings and the number of available embedders to evaluate. the $\overline{\mathcal{I}_S}$ score can capture many different aspects of the embeddings, such as the quality of the embeddings. We provide in this section a comprehensive set of ablation studies to evaluate the impact of these different factors on the $\overline{\mathcal{I}_S}$ score of the embeddings.

Table 3: Summary of the evaluated embedders with their performance on the MTEB benchmark.

| Model | Average | Classification | Clustering | Reranking | Retrieval | STS |
|---|---|---|---|---|---|---|
| SFR-Embedding-Mistral | 67.56 | 78.33 | 51.67 | 60.64 | 59.00 | 85.05 |
| echo-mistral-7b-instruct-lasttoken | 64.68 | 77.43 | 46.32 | 58.14 | 55.52 | 82.56 |
| stella-base-en-v2 | 62.61 | 75.28 | 44.90 | 58.78 | 50.10 | 83.02 |
| e5-large-v2 | 62.25 | 75.24 | 44.49 | 56.61 | 50.56 | 82.05 |
| GritLM-7B | 66.76 | 79.46 | 50.61 | 60.49 | 57.41 | 83.35 |
| ember-v1 | 63.54 | 75.99 | 45.58 | 60.04 | 51.92 | 83.34 |
| gte-large | 63.13 | 73.33 | 46.84 | 59.13 | 52.22 | 83.35 |
| UAE-Large-V1 | 64.64 | 75.58 | 46.73 | 59.88 | 54.66 | 84.54 |
| gte-base | 62.39 | 73.01 | 46.20 | 58.61 | 51.14 | 82.30 |
| sf_model_e5 | 63.34 | 73.96 | 46.61 | 59.86 | 51.80 | 83.85 |
| GIST-Embedding-v0 | 63.71 | 76.03 | 46.21 | 59.37 | 52.31 | 83.51 |
| gtr-t5-large | 58.28 | 67.14 | 41.60 | 55.36 | 47.42 | 78.19 |
| gtr-t5-xl | 58.42 | 67.11 | 41.51 | 55.96 | 47.96 | 77.80 |
| bge-base-en-v1.5 | 63.55 | 75.53 | 45.77 | 58.86 | 53.25 | 82.40 |
| sentence-t5-large | 57.06 | 72.31 | 41.65 | 54.00 | 36.71 | 81.83 |
| gemma-7b-it | N/A | N/A | N/A | N/A | N/A | N/A |
| gte-tiny | 58.69 | 70.35 | 42.09 | 55.77 | 44.92 | 80.46 |
| gtr-t5-base | 56.19 | 65.25 | 38.63 | 54.23 | 44.67 | 77.07 |
| Llama-2-7b-hf | N/A | N/A | N/A | N/A | N/A | N/A |
| sentence-t5-xl | 57.87 | 72.84 | 42.34 | 54.71 | 38.47 | 81.66 |
| gemma-2b-it | N/A | N/A | N/A | N/A | N/A | N/A |
| e5-small | 58.89 | 71.67 | 39.51 | 54.45 | 46.01 | 80.87 |
| bge-micro-v2 | 56.57 | 68.04 | 39.18 | 54.29 | 42.56 | 78.65 |
| all-distilroberta-v1 | N/A | N/A | N/A | N/A | N/A | N/A |
| multilingual-e5-small | 57.87 | 70.74 | 37.08 | 53.87 | 46.64 | 79.10 |
| msmarco-bert-co-condensor | 52.35 | 64.71 | 37.64 | 51.84 | 32.96 | 76.47 |
| sup-simcse-bert-base-uncased | 48.87 | 67.32 | 33.43 | 47.54 | 21.82 | 79.12 |
| all-MiniLM-L6-v2 | 56.26 | 63.05 | 42.35 | 58.04 | 41.95 | 78.90 |
| all-mpnet-base-v2 | 57.78 | 65.07 | 43.69 | 59.36 | 43.81 | 80.28 |
| udever-bloom-560m | 55.81 | 68.04 | 36.89 | 52.60 | 41.19 | 79.93 |
| LaBSE | 45.21 | 62.71 | 29.55 | 48.42 | 18.99 | 70.80 |
| average_word_embeddings_komninos | 42.06 | 57.65 | 26.57 | 44.75 | 21.22 | 62.46 |
| average_word_embeddings_glove.6B.300d | 41.96 | 57.29 | 27.73 | 43.29 | 21.62 | 61.85 |
| allenai-specter | 40.28 | 52.37 | 34.06 | 48.10 | 15.88 | 61.02 |

**Impact of instruction finetuning.** Instruction finetuning is now a common practice to improve the alignment of the base of models and expand the models' reasoning capabilities. In Figure 4c, show that instruction fine-tuning positively impacts the models' performance on the downstream tasks and that the $\overline{\mathcal{I}_S}$ score captures this improvement. In addition to studying the impact of instruction finetuning, we evaluated models at different checkpoints during their initial pretraining in Sec. C.3.3 using the CroissantLLM checkpoints [37].

### C.3.3 Impact of training steps

Surprisingly, we found that the number of training steps does not significantly impact the models' performance on the downstream tasks nor on the $\overline{\mathcal{I}_S}$ score of the embeddings. The $\overline{\mathcal{I}_S}$ score correctly captures this behavior as shown in Figure 7. We hypothesize that the $\overline{\mathcal{I}_S}$ score in terms of embeddings is, in this case, determined by a few numbers of training steps (the first 5000) and the overall architecture of the model. Training the model further even leads to a decrease in performance on the downstream tasks, which is not captured by the $\overline{\mathcal{I}_S}$ score of the embeddings; this could be due to the very small variation in the performance.

### C.3.4 Importance of embedding size normalization

We found that considering the amount of information packed by an embedding per coordinate is crucial to obtain a good ranking of the models. In Figure 16b, we show the correlation between the performance of the models on the MTEB benchmark and their $\overline{\mathcal{I}_S}$ score, not normalized by embedding size. While positive significative correlation is still present, the correlation is much weaker than when the dimension of the embeddings normalizes the information sufficiency.

| Task category | | $\varrho_p$ | $\varrho_s$ | $\tau$ |
|---|---|---|---|---|
| | **Average** | **0.90** | **0.83** | **0.68** |
| | AmazonCounterfactualClassification (en) | 0.75 | 0.67 | 0.50 |
| | AmazonPolarityClassification | 0.78 | 0.77 | 0.58 |
| | AmazonReviewsClassification (en) | 0.83 | 0.82 | 0.65 |
| | Banking77Classification | 0.92 | 0.84 | 0.65 |
| | EmotionClassification | 0.87 | 0.76 | 0.58 |
| Classification | ImdbClassification | 0.77 | 0.76 | 0.57 |
| | MassiveIntentClassification (en) | 0.93 | 0.84 | 0.68 |
| | MassiveScenarioClassification (en) | 0.92 | 0.84 | 0.67 |
| | MTOPDomainClassification (en) | 0.94 | 0.89 | 0.72 |
| | MTOPIntentClassification (en) | 0.80 | 0.76 | 0.59 |
| | ToxicConversationsClassification | 0.68 | 0.51 | 0.36 |
| | TweetSentimentExtractionClassification | 0.70 | 0.50 | 0.35 |
| | **Average** | **0.88** | **0.82** | **0.63** |
| | ArxivClusteringP2P | 0.59 | 0.64 | 0.46 |
| | ArxivClusteringS2S | 0.66 | 0.73 | 0.54 |
| | BiorxivClusteringP2P | 0.43 | 0.39 | 0.30 |
| | BiorxivClusteringS2S | 0.58 | 0.64 | 0.42 |
| | MedrxivClusteringP2P | 0.41 | 0.34 | 0.26 |
| Clustering | MedrxivClusteringS2S | 0.58 | 0.48 | 0.34 |
| | RedditClustering | 0.92 | 0.84 | 0.67 |
| | RedditClusteringP2P | 0.86 | 0.88 | 0.72 |
| | StackExchangeClustering | 0.93 | 0.88 | 0.71 |
| | StackExchangeClusteringP2P | 0.66 | 0.63 | 0.46 |
| | TwentyNewsgroupsClustering | 0.91 | 0.86 | 0.68 |
| | **Average** | **0.91** | **0.83** | **0.67** |
| PairClassification | SprintDuplicateQuestions | 0.69 | 0.63 | 0.44 |
| | TwitterSemEval2015 | 0.92 | 0.76 | 0.57 |
| | TwitterURLCorpus | 0.85 | 0.81 | 0.64 |
| | **Average** | **0.85** | **0.74** | **0.58** |
| | AskUbuntuDupQuestions | 0.85 | 0.65 | 0.53 |
| Reranking | MindSmallReranking | 0.86 | 0.84 | 0.65 |
| | SciDocsRR | 0.63 | 0.50 | 0.35 |
| | StackOverflowDupQuestions | 0.89 | 0.77 | 0.58 |
| | **Average** | **0.89** | **0.87** | **0.67** |
| | ArguAna | 0.80 | 0.77 | 0.59 |
| | ClimateFEVER | 0.75 | 0.78 | 0.59 |
| | CQADupstackRetrieval | 0.85 | 0.74 | 0.59 |
| | DBPedia | 0.94 | 0.90 | 0.74 |
| | FEVER | 0.87 | 0.86 | 0.67 |
| | FiQA2018 | 0.85 | 0.74 | 0.57 |
| | HotpotQA | 0.88 | 0.87 | 0.69 |
| Retrieval | MSMARCO | 0.84 | 0.78 | 0.60 |
| | NFCorpus | 0.91 | 0.86 | 0.68 |
| | NQ | 0.88 | 0.78 | 0.61 |
| | QuoraRetrieval | 0.90 | 0.80 | 0.63 |
| | SCIDOCS | 0.79 | 0.64 | 0.49 |
| | SciFact | 0.79 | 0.78 | 0.57 |
| | Touche2020 | 0.71 | 0.58 | 0.41 |
| | TRECCOVID | 0.73 | 0.65 | 0.51 |
| | **Average** | **0.90** | **0.75** | **0.57** |
| | STSBenchmark | 0.85 | 0.70 | 0.53 |
| | BIOSSES | 0.80 | 0.72 | 0.52 |
| | SICK-R | 0.83 | 0.64 | 0.46 |
| | STS12 | 0.80 | 0.63 | 0.44 |
| STS | STS13 | 0.86 | 0.66 | 0.46 |
| | STS14 | 0.89 | 0.68 | 0.47 |
| | STS15 | 0.91 | 0.79 | 0.62 |
| | STS16 | 0.91 | 0.76 | 0.59 |
| | STS17 (en-en) | 0.81 | 0.45 | 0.32 |
| | STS22 (en) | 0.84 | 0.61 | 0.45 |
| Summarization | SummEval | 0.46 | 0.31 | 0.21 |

Table 4: Detailed correlations between the $\overline{\overline{\mathcal{I}_S}}$ score of the models and their performance on the MTEB benchmark.

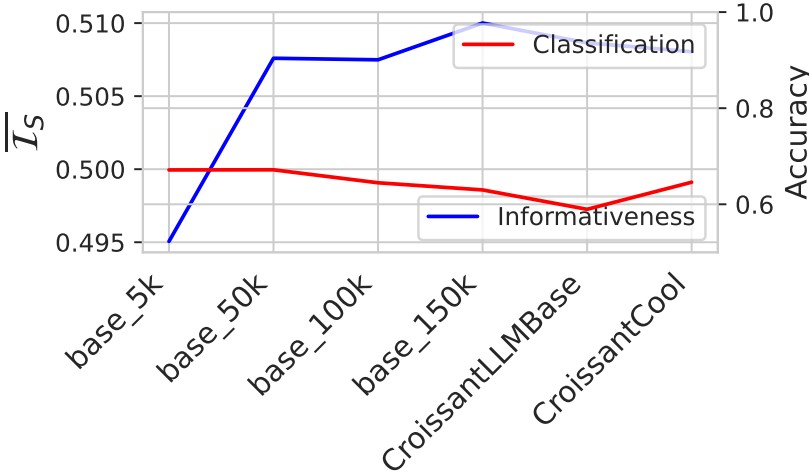

Figure 7: Impact of the number of training steps on the performance of the models on the downstream tasks and their $\overline{\mathcal{I}_S}$ score.

inforatmation sufficiency

### C.3.5 Community and cluster performance

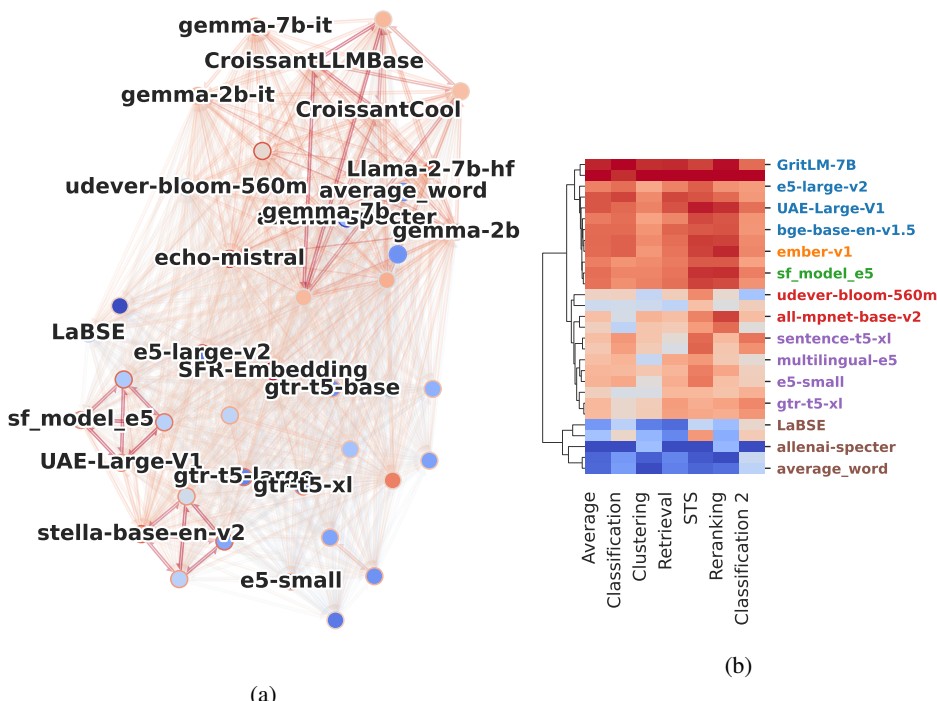

(a)

(b)

Figure 8: We present different interesting properties of $\overline{\mathcal{I}_S}$. In Figure 4a, we show that it can be used to cluster models Figure 8b, reports the performance of the models on the different task categories. They are grouped by similar behaviors on these tasks (dendrograms) and colored by the communities discovered in the information sufficiency graph. (Only models evaluated as part of the MTEB benchmark are shown).

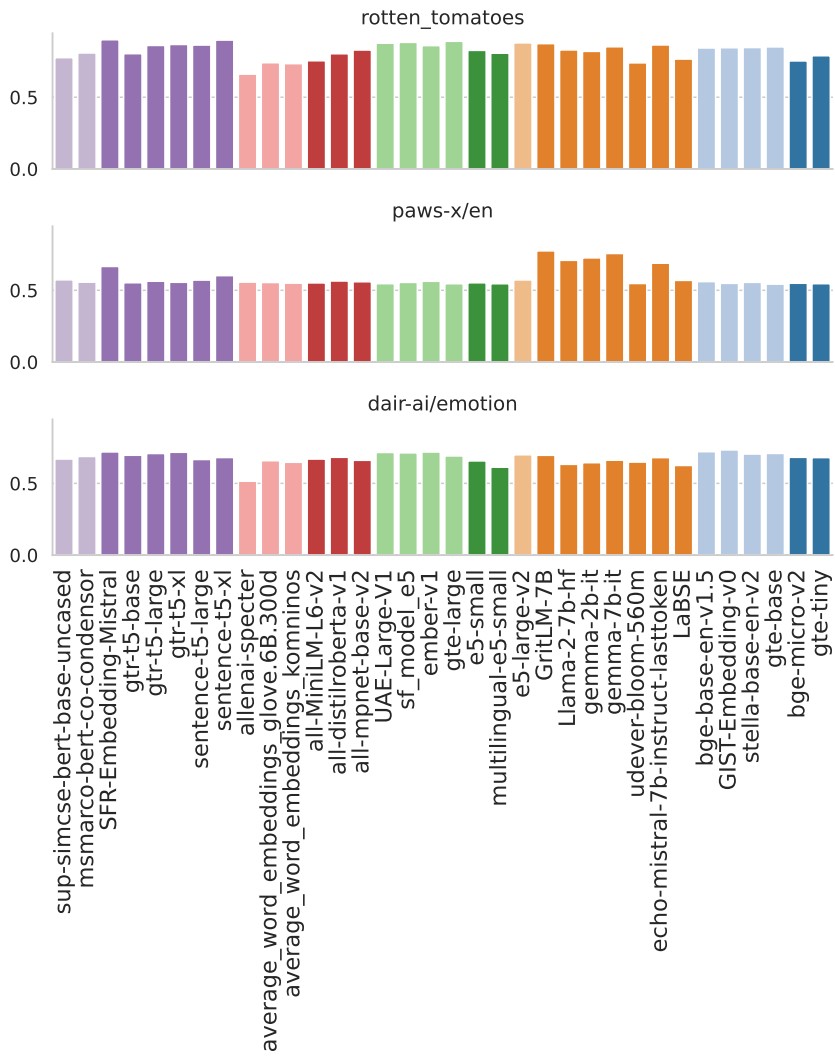

Figure 9: Performance of the models on the downstream tasks grouped by clusters discovered by the directed $\overline{\mathcal{I}_S}$.

We postulate that models clustered together by information sufficiency are likely to behave similarly on the downstream tasks. We evaluate this hypothesis by grouping the models by clusters discovered using the information sufficiency and reporting their performance on the downstream tasks. In Figure 8b and Figure 9, we observe that models within the same cluster tend to have similar behaviors on the downstream tasks.

### C.3.6 Evaluating information sufficiency on different datasets

The $\overline{\mathcal{I}_S}$ score is evaluated with a fixed dataset supposed to represent the data distribution of interest (either a very diverse set or a subset following a distribution specific to a subfield like medical or legal texts). We cross-evaluated the $\overline{\mathcal{I}_S}$ of the models on different datasets and the performance of the models on the downstream tasks in Figure 10. We find that closer datasets in terms of the data distribution lead to a higher correlation between the $\overline{\mathcal{I}_S}$ score of the models and their performance on the downstream tasks. It is especially highlighted when comparing the correlations we get when evaluating $\overline{\mathcal{I}_S}$ on the AG News and Amazon polarity datasets. The first one corresponds to news articles, and the task is to guess the topic, whereas Amazon Polarity corresponds to product reviews, which is a sentiment analysis task. We find that the $\overline{\mathcal{I}_S}$ score evaluated on Amazon Polarity tends to yield way better correlation with the performance on the sentiment analysis downstream tasks

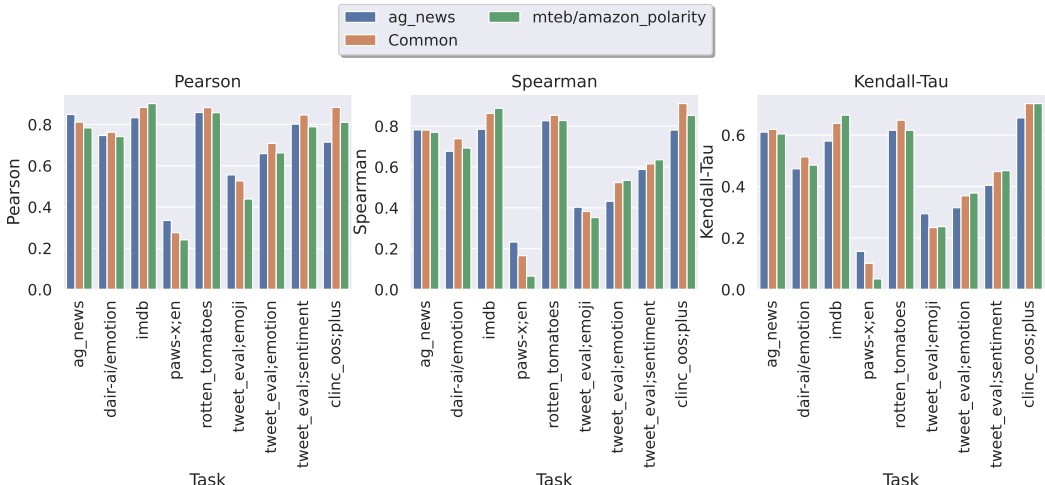

Figure 10: Correlation between $\overline{\mathcal{I}_S}$ scores computed on different datasets and the cross-performance on different tasks.

such as tweet_eval/sentiment, tweet_eval/emotion, IMDB or Rotten Tomatoes or to a lesser extent dair/emotion. Interestingly, the difference is less significant on the tweet_eval/emoji subtask.

# D    Molecular Experiment Details

## D.1    Embedders considered

| Model name | SMILES | 2D-GNN | 3D-GNN | Architecture | Out size | Dataset (size) |
|---|---|---|---|---|---|---|
| Not-trained | | ✓ | | GIN | 300 | - |
| AttributeMask[99] | | ✓ | | GIN | 300 | GEOM [6] (50k) |
| ContextPred[99] | | ✓ | | GIN | 300 | GEOM [6] (50k) |
| GPT-GNN[48] | | ✓ | | GIN | 300 | GEOM [6] (50k) |
| InfoGraph[98] | | ✓ | | GIN | 300 | GEOM [6] (50k) |
| GraphCL[119] | | ✓ | | GIN | 300 | GEOM [6] (50k) |
| GROVER[89] | | ✓ | | GIN | 300 | GEOM [6] (50k) |
| GraphLog[116] | | ✓ | | GIN | 300 | GEOM [6] (50k) |
| GraphMVP[67][1] | | ✓ | | GIN | 300 | GEOM [6] (50k) |
| 3D-infomax[96][1] | | ✓ | | PNA | 800 | QMugs [51] (620k) |
| MolR[112] | | ✓ | | GCN, GAT, TAG | 1024 | USPTO[112] ( 1.5M) |
| MoleOOD[117] | | ✓ | | GIN , SAGE, GCN | 256 | BACE [47] (400k) |
| ChemBERT MLM[3] | ✓ | | | RoBERTa | 600 | PubChem [55] (5M, 10M, 77M) |
| ChemBERT MTR[3] | ✓ | | | RoBERTa | 384 | PubChem [55] (5M, 10M, 77M) |
| MolBert[36] | ✓ | | | BERT | 256 | GuacaMol [18] (1.6M) |
| ChemGPT[40] | ✓ | | | GPT | 128, 256, 2048 | PubChem [55] (10M) |
| 3D-denosing[121] | | | ✓ | TorchMD-net | 256 | PCQM4Mv2[46] (3.7M) |
| 3D-fractional[39] | | | ✓ | TorchMD-net | 256 | PCQM4Mv2[46] (3.7M) |

Table 5: Models evaluated on the ZINC dataset.

We considered 28 models for the molecular experiments, summed up in Tab. 5. Some models were used in different versions (architectures, number of parameters, pretraining dataset's size), such as the ChemBert models, followed by the size of their datasets, or ChemGPT, followed by their number of parameters.

Most 2D-GNNs were trained on the GEOM [6] dataset and were gathered from the repository of GraphMVP [67] model. Note that the MoleOOD [117] model was trained on the BACE [47] dataset, with a supervised task specific to the $\beta$-secretase enzyme. As a result, this model can be seen as "already specialized", explaining its poor performance in our evaluation.

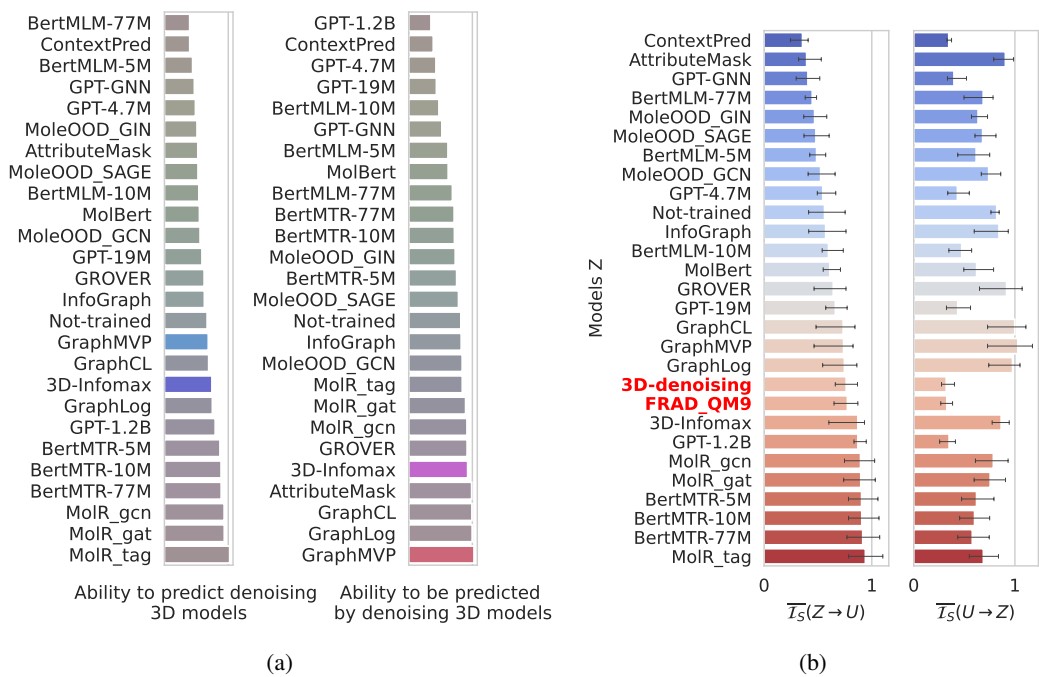

Figure 11: (a) Information sufficiency of embedders over 3D-Denoising models (left) and of 3D-Denoising models over the other embedders (right). (b) Information sufficiency of embedders in both directions. We see the 3D denoising models are among the least predicted models.

We used the RD-Kit and Datamol tool-kits[61, 71] to pre-process the molecules and to generate three-dimensional conformers for 3D-models. To run the models using 3D views of the molecules, we generated five conformers (possible 3D configuration of the molecule) for each SMILES and kept the conformer with the lowest energy. Note that this methodology is imperfect, as the 3D coordinates might be noisy; however, we followed the same procedure to pre-process the ZINC dataset, to evaluate the information sufficiency, and on the datasets corresponding to each downstream task.

Finally we considered a variety of models architecture for 2D-GNNs notably graph isomorphism network (GIN) [115], principal neighbor aggregation networks (PNA) [26], graph convolutional network (GCN) [33], graph attention network (GAT) [110], topology adaptive graph convolutional networks (TAG) [32] and GraphSAGE [44]. For SMILES-based models, backbones are inspired by BERT [31], RoBERTa [69], and GPT [40]. Finally, both our 3D models use TorchMD-net[80] as a backbone.

### D.2  Details on the information sufficiency estimation

**3D models.** The two 3D models considered (FRAD [39] and Denoising [121]) obtain high $\overline{\mathcal{I}_S}$ scores while being among the least predictable (Figure 11b and Figure 12). This suggests that these models capture 3D-specific features inaccessible from other modalities while maintaining sufficient overlap to predict them.

**2D-3D models** Some 2D-GNNs we considered (GraphMVP and 3D-infomax) are trained to maximize the mutual information between their embeddings and 3D representations of the molecule. Hence, we expect these models to be related to the 3D-denoising models we considered. However, we observe in Figure 11a that these models do not achieve particularly high information sufficiency scores over 3D-denoising models. On the other hand, the 3D models achieve high information sufficiency scores over them, which might suggest that these 2D models and 3D-denoising models share information that is easier to access from the 3D models. However, we want to point out that GraphMVP and 3D-infomax are both among the most predicted models; that is to say, among the other models in our pool, they achieve the highest information sufficiency scores.

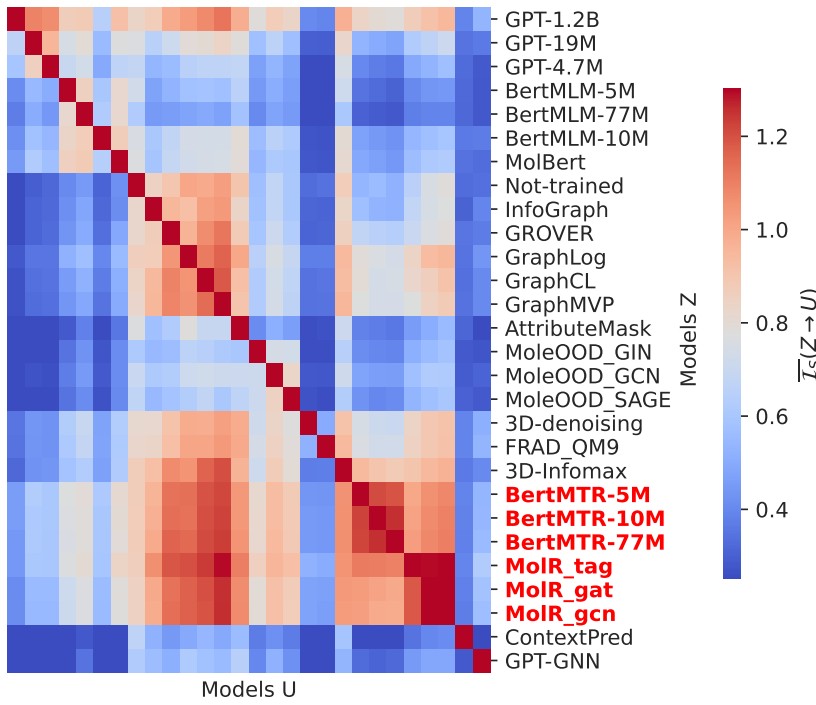

Figure 12: Pairwise information sufficiency between molecular embedders.

## D.3 Complementary results on ADMET tasks

The datasets chosen for the molecular experiments are extracted from the Therapeutics Data Commons (TDC) [49] platform. We focused our experiments on ADMET tasks, crucial for drug discovery and development, which results in a total of 31 tasks, described in Tab. 6.

Each dataset is split into a training set, a validation set, and a test set following a scaffold split. This splitting strategy ensures that molecules sharing a similar scaffold will be part of the same split in the task. This corresponds to a more realistic scenario, where practitioners only have access

| Dropout rate | hidden dimension | network depth | n-epochs |
|---|---|---|---|
| 0, 0.2 | 4, 8, 16, 32, 64, 128 | 1, 2, 3 | $\min(200, 200 * \frac{5000}{\text{task size}})$ |

Table 7: Hyperparameters tuned for the evaluation of embedders on ADMET downstream tasks

to molecules belonging to the same chemical series. Classifiers are then trained on the training set of each task, where the best hyperparameters and checkpoints are selected on the validation set. The final performance is finally measured on the test set, and we run each experiment 10 times with different random seeds.

A grid search is performed on each dataset individually to maximize the average AUROC or $R^2$ score across all models for binary classification and regression. We chose a maximum number of epochs depending on the task size to ensure all models have time to converge, limiting this amount to grow to at most 200 epochs.

Tab. 6 also displays the variation of the correlation coefficient between the ranking obtained on the $\overline{\mathcal{I}_S}$ score and the performances obtained on the downstream tasks regarding Spearman and Kendall correlations. We can see that the $\overline{\mathcal{I}_S}$ score correlates well with the performance of downstream tasks when the amount of data available is large.

Finally, we can see in Figure 13 that by grouping models based on their performances on these tasks, we obtain a similar clustering to the one obtained on the $\overline{\mathcal{I}_S}$ score in Sec. 6.2, with NLP-inspired models grouped. Similarly, the *tinyChem*Bert-MTR and MolR models are also grouped.

| Category | Model name | Task size | cls | reg | Correlation $\varrho_p$ | $\varrho_s$ | $\tau$ | Avg. metric in the grid |
|---|---|---|---|---|---|---|---|---|
| Absorption | P-glycoprotein Inhibition | 1212 | ✓ | | 0.92 | 0.93 | 0.76 | $0.88 \pm 0.03$ |
| | AqSolDB | 9982 | | ✓ | 0.91 | 0.91 | 0.75 | $0.52 \pm 0.09$ |
| | Lipophilicity | 4200 | | ✓ | 0.88 | 0.89 | 0.71 | $0.29 \pm 0.07$ |
| | Caco-2 Permeability | 906 | | ✓ | 0.77 | 0.80 | 0.61 | $0.32 \pm 0.08$ |
| | Human Intestinal Absorption | 578 | ✓ | | 0.79 | 0.77 | 0.54 | $0.67 \pm 0.02$ |
| | FreeSolv | 642 | | ✓ | 0.65 | 0.73 | 0.53 | $0.36 \pm 0.12$ |
| | PAMPA Permeability | 2035 | ✓ | | 0.61 | 0.63 | 0.44 | $0.67 \pm 0.02$ |
| | Oral Bioavailability | 640 | ✓ | | 0.50 | 0.45 | 0.33 | $0.68 \pm 0.02$ |
| Distribution | Plasma-Protein BDR | 1614 | | ✓ | 0.85 | 0.84 | 0.68 | $0.18 \pm 0.04$ |
| | Blood-Brain barrier | 1975 | ✓ | | 0.79 | 0.81 | 0.60 | $0.32 \pm 0.08$ |
| | VDss | 1130 | | ✓ | 0.71 | 0.73 | 0.53 | $0.16 \pm 0.05$ |
| Metabolism | CYPP450 3A4 Inhib. | 12328 | ✓ | | 0.96 | 0.96 | 0.85 | $0.80 \pm 0.04$ |
| | CYPP450 1A2 Inhib. | 12579 | ✓ | | 0.94 | 0.95 | 0.81 | $0.87 \pm 0.03$ |
| | CYPP450 2C19 Inhib. | 12665 | ✓ | | 0.94 | 0.95 | 0.75 | $0.82 \pm 0.03$ |
| | CYPP450 2C9 Inhib. | 12092 | ✓ | | 0.92 | 0.92 | 0.79 | $0.83 \pm 0.03$ |
| | CYPP450 2D6 Inhib. | 13130 | ✓ | | 0.94 | 0.91 | 0.74 | $0.78 \pm 0.04$ |
| | CYPP450 2D6 Substrate | 664 | ✓ | | 0.75 | 0.74 | 0.57 | $0.75 \pm 0.03$ |
| | CYPP450 3A4 Substrate | 667 | ✓ | | 0.50 | 0.53 | 0.35 | $0.63 \pm 0.02$ |
| | CYPP450 2C9 Substrate | 666 | ✓ | | 0.20 | 0.13 | 0.09 | $0.65 \pm 0.02$ |
| Excretion | Clearance hepatocyte | 1020 | | ✓ | 0.78 | 0.79 | 0.57 | $0.11 \pm 0.03$ |
| | Half Life | 667 | | ✓ | 0.76 | 0.78 | 0.58 | $-0.06 \pm 0.11$ |
| | Clearance microsome | 1102 | | ✓ | 0.72 | 0.72 | 0.54 | $0.08 \pm 0.03$ |
| Toxicity | Tox21 | 7831 | ✓ | | 0.93 | 0.93 | 0.78 | $0.75 \pm 0.03$ |
| | hERG | 13445 | ✓ | | 0.91 | 0.90 | 0.75 | $0.76 \pm 0.04$ |
| | | 648 | ✓ | | 0.81 | 0.84 | 0.63 | $0.75 \pm 0.03$ |
| | Acute Toxicity LD50 | 7385 | | ✓ | 0.82 | 0.82 | 0.63 | $0.16 \pm 0.04$ |
| | Ames Mutagenicity | 7255 | ✓ | | 0.79 | 0.78 | 0.60 | $0.79 \pm 0.03$ |
| | ClinTox | 1484 | ✓ | | 0.69 | 0.69 | 0.49 | $0.71 \pm 0.03$ |
| | Carcinogens | 278 | ✓ | | 0.47 | 0.49 | 0.35 | $0.76 \pm 0.08$ |
| | Drug Induced Liver Injury | 475 | ✓ | | 0.39 | 0.36 | 0.25 | $0.83 \pm 0.03$ |
| | Skin Reaction | 404 | ✓ | | 0.01 | 0.07 | 0.06 | $0.64 \pm 0.03$ |

Table 6: ADMET tasks extracted from the Therapeutic Data Commons platform [49] considered in our experiments. We report the correlation between the informativeness score and the performances of the embedders on the downstream tasks in terms of Pearson correlation $\varrho_p$, Spearman correlation $\varrho_s$ and Kendall-Tau $\tau$. We also report the average metric of the models on each task across the grid search runs, in terms of $R^2$ **for regression tasks** and **AUROC for classification tasks**. The tasks are ordered within each category by the correlation with the informativness score (in terms of Spearman correlation).

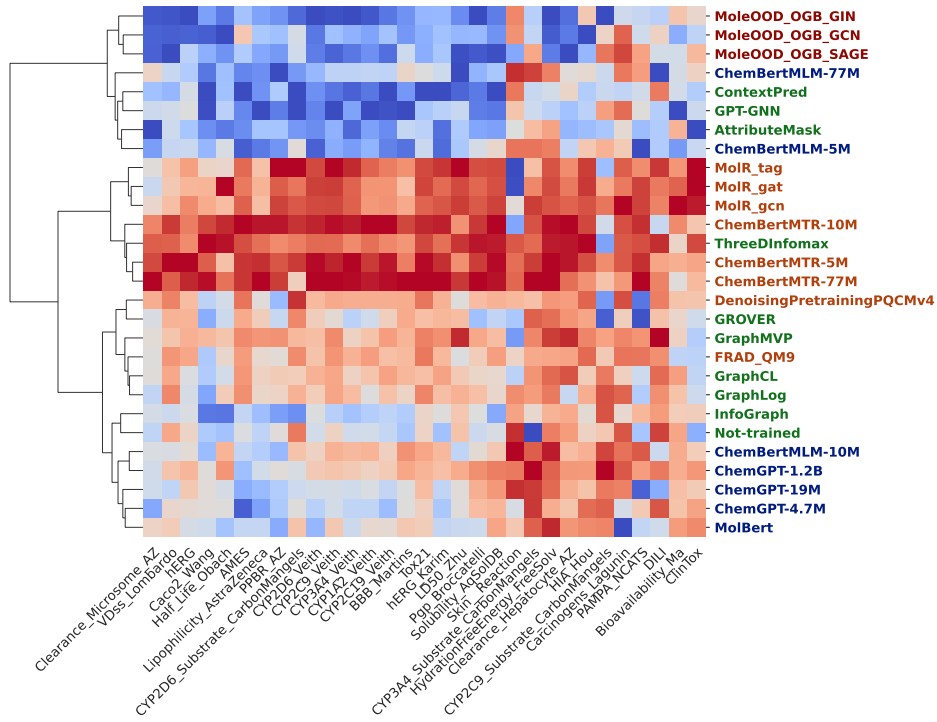

Figure 13: Heatmap representation of the performances of the different models on the downstream tasks, where embedders behaving similarly on the various tasks are clustered, and the embedders are colored based on their community computed in Sec. 6.2 based on the $\overline{\mathcal{I}_S}$ score.

### D.4 Drug target Interaction prediction

We propose further evaluating the embedders on yet another type of downstream task: Drug-Target Interaction. This task aims to predict the binding affinity between a given pair (drug, target). Since none of our models can process protein sequences, we decompose each dataset into multiple regression tasks on a single target by querying all molecules associated with a label for this target. Each task is then formulated as a set of molecules: $\mathcal{X} = \{x_i\}_{i \in \{0, ..., N\}}$, and their labels $\mathcal{Y} = \{y_i\}_{i \in \{0, ..., N\}}$

However, such tasks can be small, making it hard to build proper models from the embeddings. In contrast, the number of tasks is very large, making it computationally expensive to proceed to

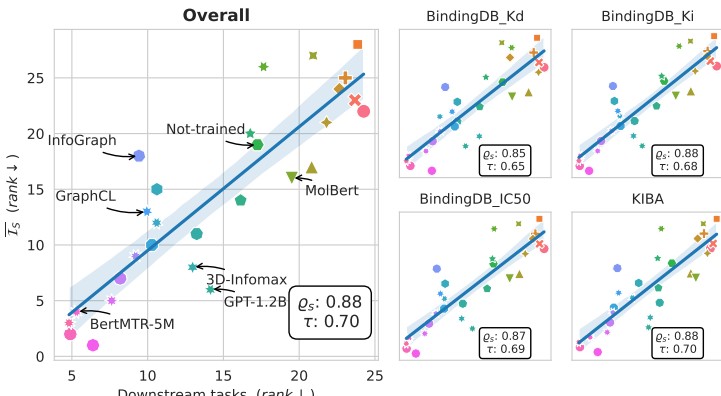

Figure 14: Correlations of the $\overline{\mathcal{I}_S}$ score with the performances on the DTI tasks, in terms of Spearman and Kendall coefficients.

an adequate hyperparameter selection. To bypass these limitations, we propose to estimate the embedded space's clustering quality for each model by measuring how close the labels of a molecule are compared to its nearest neighbors for each task. In other words, we measure:

$$\tilde{\mathcal{L}}_{n_{\text{neigh}}}(\mathcal{X}, \mathcal{Y}) = \frac{1}{N} \sum_{i=0}^{N} \mathcal{L}_{n_{\text{neigh}}}(i, \mathcal{X}, \mathcal{Y}),$$

$$\text{with} \quad \mathcal{L}_{n_{\text{neigh}}}(i, \mathcal{X}, \mathcal{Y}) = \frac{1}{n_{\text{neigh}}} \sum_{j \in \mathcal{N}_{i, n_{\text{neigh}}}(\mathcal{X})} \|y_i - y_j\|^2, \tag{26}$$

and $\mathcal{N}_{i, n_{\text{neigh}}}(\mathcal{X})$ is the set containing the $n_{\text{neigh}}$ closest neighbors of $x_i \in \mathcal{X}$, which would be the performances of a K-nearest neighbors regressor on the task when using one data sample.

This quantity can be interpreted as a proxy of the embedding's capability to perform a similarity search, a classic chemo-informatic method using the similarity between different molecular projections to perform predictions. This training-free and computationally inexpensive approach allows us to evaluate the models on many tasks/targets.

We focused on 4 DTI datasets: KIBA [101], BindingDB-Kd, BindingDB-Ki, and BindingDB-IC50 [68], with a total of $1496$ tasks. We removed all tasks containing less than 128 molecules to ensure minimum data for the clustering evaluation.

| | KIBA (175 targets) | | | | BindingDB-Kd (22 targets) | | | | BindingDB-Ki (372 targets) | | | | BindingDB-IC50 (927 targets) | | | |
|---|---|---|---|---|---|---|---|---|---|---|---|---|---|---|---|---|
| $n_{neighb}$ | 1 | 2 | 4 | 8 | 1 | 2 | 4 | 8 | 1 | 2 | 4 | 8 | 1 | 2 | 4 | 8 |
| $\varrho_p$ | 0.85 | 0.84 | 0.84 | 0.83 | 0.90 | 0.90 | 0.87 | 0.84 | 0.88 | 0.88 | 0.88 | 0.87 | 0.87 | 0.86 | 0.86 | 0.86 |
| $\varrho_s$ | 0.88 | 0.88 | 0.87 | 0.86 | 0.87 | 0.86 | 0.85 | 0.82 | 0.87 | 0.87 | 0.87 | 0.88 | 0.86 | 0.87 | 0.87 | 0.86 |
| $\tau$ | 0.72 | 0.71 | 0.69 | 0.67 | 0.68 | 0.66 | 0.65 | 0.62 | 0.68 | 0.67 | 0.69 | 0.69 | 0.66 | 0.68 | 0.69 | 0.68 |

Table 8: Correlation between $\overline{\mathcal{I}_S}$'s informativness score and our clustering evaluation score $\tilde{\mathcal{L}}_{n_{\text{neighb}}}$ on the four DTI datasets considered.

We obtain similar results as in Sec. 6.1, our metric correlating with the performances on the different tasks considered. Figure 14 sums up all results by establishing a ranking across models and the number of neighbors, where we can see that MolR and ChemBerta-MTR appear as both the most promising models according to their $\overline{\mathcal{I}_S}$ score, and the best models evaluated. Furthermore, the outliers observed inSec. 6.1 show different behaviors in this setting. For instance, while 3D-Infomax seemed under-estimated and InfoGraph over-estimated by the $\overline{\mathcal{I}_S}$ score after seeing the results on the ADMET tasks, Infograph appears under-estimated in this setup, and 3D-infomax over-estimated.

# E   Information Sufficiency Estimation

## E.1   Estimation method

As stated in Sec. 2, the deficiency $\delta(P_{U|X} \to P_{Z|X})$ is an intractable object measuring the cost of the reconstruction of $Z$ from $U$. Due to this intractability, we propose to estimate the information sufficiency $\mathcal{I}_S(U \to Z)$, which is a tractable proxy for the deficiency.

### E.1.1   KNIFE Estimator

We recall the definition of the information sufficiency:

$$\mathcal{I}_S(U \to Z) \triangleq \underbrace{\inf_{f \in \mathcal{F}_\Theta(\mathsf{Z})} \mathbb{E}\left[-\log f(Z)\right]}_{\text{Uncertainty of } Z} - \underbrace{\mathbb{E}\left[\inf_{M \in \mathcal{K}_\Theta(\mathsf{Z}|\mathsf{U})} \mathbb{E}\left[-\log M(Z|U)|U\right]\right]}_{\text{Uncertainty when simulating } Z \text{ from } U \text{ with } M}. \tag{27}$$

We denote $C$ the number of modes chosen for the Gaussian mixture distributions and $\text{GM}_{\boldsymbol{\mu}, \boldsymbol{\Sigma}, \boldsymbol{w}}$ the Gaussian mixture distribution with $C$ components, parametrized by $\boldsymbol{\mu}, \boldsymbol{\Sigma}, \boldsymbol{w}$, with $\mathbf{1}^T \boldsymbol{w} = 1$, such that

$\mathrm{GM}_{\boldsymbol{\mu},\boldsymbol{\Sigma},\boldsymbol{w}}(z) = \sum_{c=1}^{C} \boldsymbol{w}_c \mathcal{N}(z|\boldsymbol{\mu}_c, \boldsymbol{\Sigma}_c)$, where $\mathbf{w}_c$ is the weight of the $c$-th component, and $\mathcal{N}(z|\boldsymbol{\mu}_c, \boldsymbol{\Sigma}_c)$ is the density of a multivariate Gaussian distribution with mean $\boldsymbol{\mu}_c$ and covariance $\boldsymbol{\Sigma}_c$ the c-th mean and covariance matrix.

To estimate the information sufficiency between the two embedders $U$ and $Z$, we follow the procedure described in KNIFE [82].

$\mathcal{F}_{\Theta}$ is hence the class of multivariate Gaussian mixtures with $C$ components, it is parametrized by $\boldsymbol{\mu}, \boldsymbol{\Sigma}, \boldsymbol{w}$. These learnable parameters are optimized to maximize the log-likelihood of $Z$.[8]

The class of Markov kernels $\mathcal{K}_{\Theta}(\mathsf{Z}|\mathsf{U})$ is also composed of multivariate Gaussian mixtures whose parameters are estimated using a small feedforward neural network. Such that for each $u \in \mathsf{U}$, the parameters of the Gaussian mixture are $\boldsymbol{\mu}(u), \boldsymbol{\Sigma}(u), \boldsymbol{w}(u)$.

In practice we considered the covariance matrix to be diagonal to avoid the number of parameters of the Gaussian mixtures to grow too large with the dimension of the embeddings $d$. The number of parameters to be estimated for the Gaussian mixtures are hence: $C \times (2d + 1)$, with the number of parameters of the feedforward networks for the Markov kernels.

---

**Procedure 1** Estimation of $\mathcal{I}_S(U \to Z)$, $\mathrm{GM}_{\mu,\boldsymbol{\Sigma},\mathbf{w}}$ denotes the Gaussian Mixture model with means $\mu$, covariances $\boldsymbol{\Sigma}$ and weights $\mathbf{w}$.

---

**Input:** Pairs of corresponding embeddings $(z_i, u_i)_N$
**Output:** Information sufficiency $\mathcal{I}_S(U \to Z)$

$\quad \mu_{\mathbf{Z}}, \boldsymbol{\Sigma}_{\mathbf{Z}}, \mathbf{w}_{\mathbf{Z}} \leftarrow \arg\min_{\mu,\boldsymbol{\Sigma},\mathbf{w}} - \sum_{i=1}^{N} \log \mathrm{GM}_{\mu,\boldsymbol{\Sigma},\mathbf{w}}(z_i)$
$\quad \mu_{\mathbf{Z}|\mathbf{U}}, \boldsymbol{\Sigma}_{\mathbf{Z}|\mathbf{U}}, \mathbf{w}_{\mathbf{Z}|\mathbf{U}} \leftarrow \arg\min_{\mu,\boldsymbol{\Sigma},\mathbf{w}} - \sum_{i=1}^{N} log \, \mathrm{GM}_{\mu(u_i),\boldsymbol{\Sigma}(u_i),\mathbf{w}(u_i)}(z_i)$
$\quad$ Uncertainty of $Z$: $H(Z) \leftarrow \frac{1}{N} \sum_{i=1}^{N} \log \mathrm{GM}_{\mu_{\mathbf{Z}},\boldsymbol{\Sigma}_{\mathbf{Z}},\mathbf{w}_{\mathbf{Z}}}(z_i)$
$\quad$ Uncertainty of $Z$ given $U$: $H(Z|U) \leftarrow \frac{1}{N} \sum_{i=1}^{N} \log \mathrm{GM}_{\mu_{\mathbf{Z}|\mathbf{U}}(u_i),\boldsymbol{\Sigma}_{\mathbf{Z}|\mathbf{U}}(u_i),\mathbf{w}_{\mathbf{Z}|\mathbf{U}}(u_i)}(z_i)$
$\quad$ Return $\mathcal{I}_S(U \to Z) \leftarrow H(Z) - H(Z|U)$

---

Both the parameters of the marginal distribution of $Z$ and the parameters of the conditional distribution of $Z$ given $U$ are estimated through likelihood maximization. The uncertainty of $Z$ is estimated by the negative log-likelihood of the data under the marginal distribution of $Z$. The uncertainty of $Z$ given $U$ is estimated by the negative log-likelihood of the data under the conditional distribution of $Z$ given $U$.

### E.1.2 Embedding dimension normalization.

However, this approach faces one major drawback: it favors models that generate embeddings of high dimensionality. To evaluate the information sufficiency between the models, we estimate the uncertainty of $Z$ and the uncertainty of $Z$ given $U$. As seen in Figure 15, the estimated information sufficiency is highly correlated to the dimension of the latent space of $Z$, favoring models with high-dimensional latent spaces. This can be explained by the fact that these embedders yield larger marginal uncertainties. The resulting difference in the uncertainties $\mathcal{I}_S(U \to Z)$ is hence larger in absolute values.

We can see in Figure 15 that the dimension of the latent space of $Z$ and the uncertainty of $Z$ are evolving linearly. We thereby divide the information sufficiency by $dim(Z)$, which can be seen as an approximation of the normalization by the uncertainty of $Z$. We report in Figure 16b results without this normalization. It still correlates significantly with the downstream tasks performance, but the correlation is stronger when the normalization is applied. Hence, we focus on the relative variation of the uncertainty of $Z$ explained by $U$. Note that the uncertainty of $Z$ can be negative, as it can be assimilated to a differential entropy. As a result, the "true" relative variation of the uncertainty would not be suitable for comparing different models (as it would not guarantee any ordering). While there is a general trend where larger models do have larger embeddings and perform better, well-trained smaller embeddings are competitive with larger embeddings, and the $\overline{\mathcal{I}_S}$ score captures this behavior (Figure 16a).

---

[8]In practice since this operation does not depend on $U$, we store the weights of this distribution, and use it for all pairs $\mathcal{I}_S(. \to Z)$.

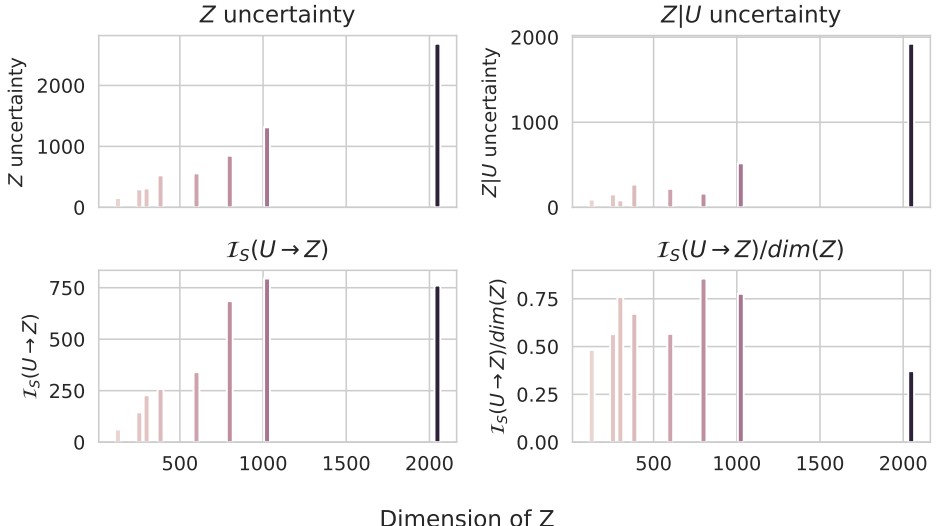

Figure 15: Relationship between the dimension of $Z$'s latent space and the quantities estimated to compute the information sufficiency in molecular modeling.

We build our proxy by measuring the median values of the set $\mathcal{S}_{\mathcal{I}_\mathcal{S}}(k) = \{\mathcal{I}_S(Z_k \to Z_l)\}_{l \neq k}$ for an embedder $Z_k$ in our pool of models.

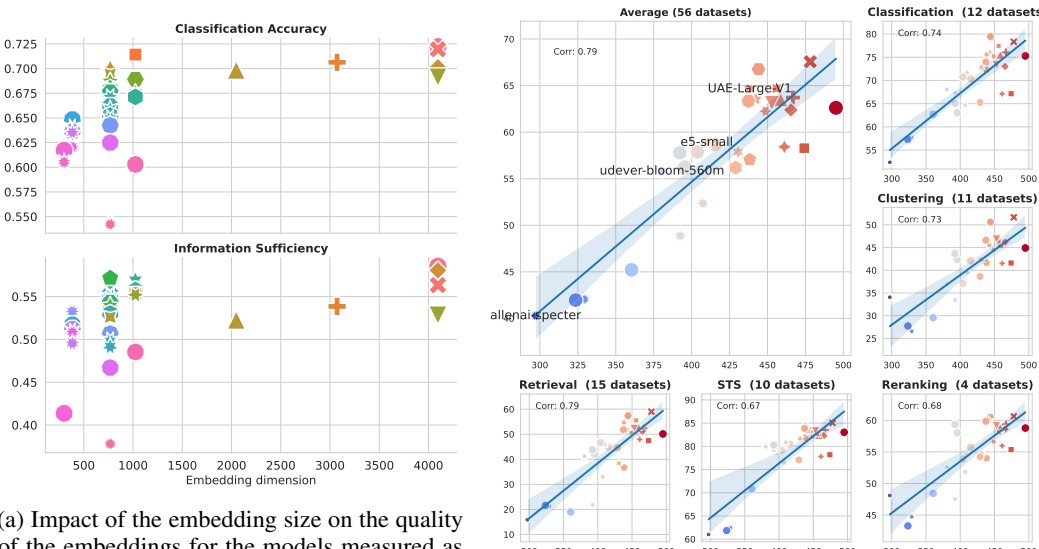

(a) Impact of the embedding size on the quality of the embeddings for the models measured as actual performance on downstream tasks and $\overline{\mathcal{I}_S}$ score.

(b) Correlations on the MTEB benchmark when the embeddings are not normalized by their size.

Figure 16

### E.1.3 Median instead of mean.

We use the median instead of the mean to compute the $\overline{\mathcal{I}_S}$ score. The median is more robust to outliers and the distribution of available embedders. For example, if many models are very similar, the mean would be biased by these models, while the median would not. Thus, we chose the median. While this change has a minor impact when there is enough diversity in the models, it can have a significant impact when the models are very similar, for example, when including different checkpoints of the same model Figure 17.

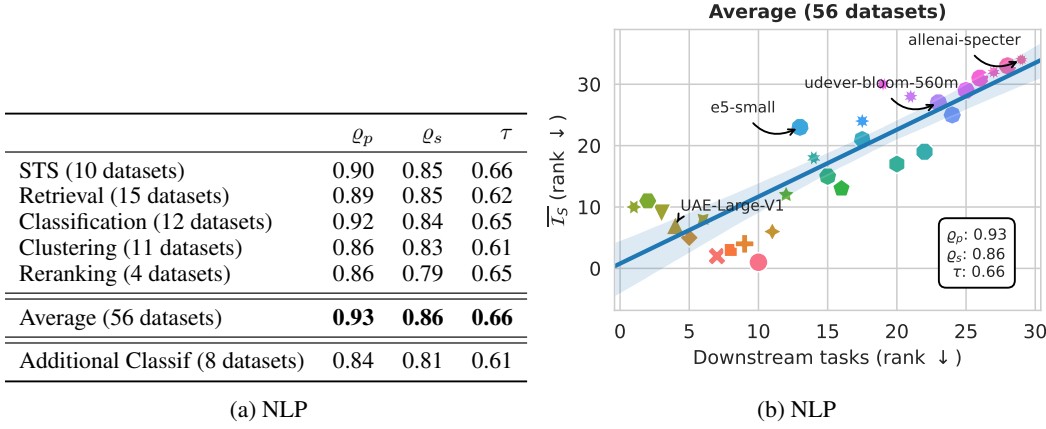

|  | $\varrho_p$ | $\varrho_s$ | $\tau$ |
|---|---|---|---|
| STS (10 datasets) | 0.90 | 0.85 | 0.66 |
| Retrieval (15 datasets) | 0.89 | 0.85 | 0.62 |
| Classification (12 datasets) | 0.92 | 0.84 | 0.65 |
| Clustering (11 datasets) | 0.86 | 0.83 | 0.61 |
| Reranking (4 datasets) | 0.86 | 0.79 | 0.65 |
| Average (56 datasets) | **0.93** | **0.86** | **0.66** |
| Additional Classif (8 datasets) | 0.84 | 0.81 | 0.61 |

(a) NLP

(b) NLP

Figure 17: Correlation between $\overline{\mathcal{I}_S}$ score computed as the mean information sufficiency and the downstream task performances in NLP. $\varrho_p$ is the Pearson correlation, $\varrho_s$ is the spearman correlation

## E.2 Hyperparameter selection

We use parametric classes composed of multivariate Gaussian mixture distributions for $\mathcal{F}_\Theta$ and $\mathcal{K}_\Theta(\mathsf{Z}|\mathsf{U})$ in the definition of the information sufficiency (Eq. 27), the number of components in the mixture is a crucial parameter that needs to be selected concerning the data distribution of interest. We ran ablation studies to evaluate the impact of the number of components in the mixture on the information sufficiency estimation and the correlation between the $\overline{\mathcal{I}_S}$ score and the downstream tasks performance (Figure 20). We found that the ideal number of components in NLP is $8$, and in molecular modeling, it is $4$. Figure 18 and Figure 19 show the embeddings of the models in the first two principal components.

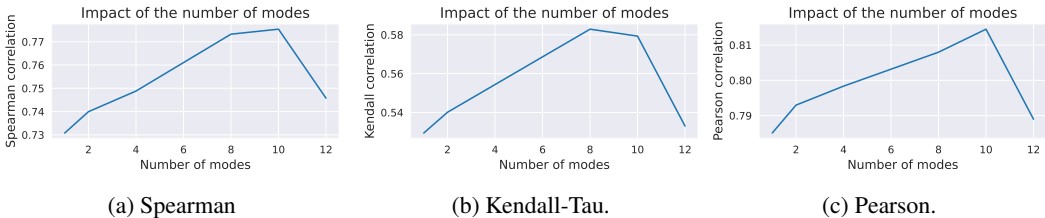

(a) Spearman

(b) Kendall-Tau.

(c) Pearson.

Figure 20: Impact of the number of modes used to estimate the $\overline{\mathcal{I}_S}$ score and its correlation with the downstream tasks performance in NLP. We chose to use $8$ modes in practice.

## E.3 Impact of the task size

Our study focused on finding the most promising model to be competitive on any downstream tasks. However, if the downstream task is not learnable, the most promising model could appear as not competitive on this specific task. In particular, if the amount of data available in the downstream task is insufficient, the differences between different embedder's representations might not be easily leveraged. This phenomenon can be seen in Figure 21, highlighting how when fewer than 1000 data points are available, the correlation between the $\overline{\mathcal{I}_S}$ score and the downstream performance becomes weaker.

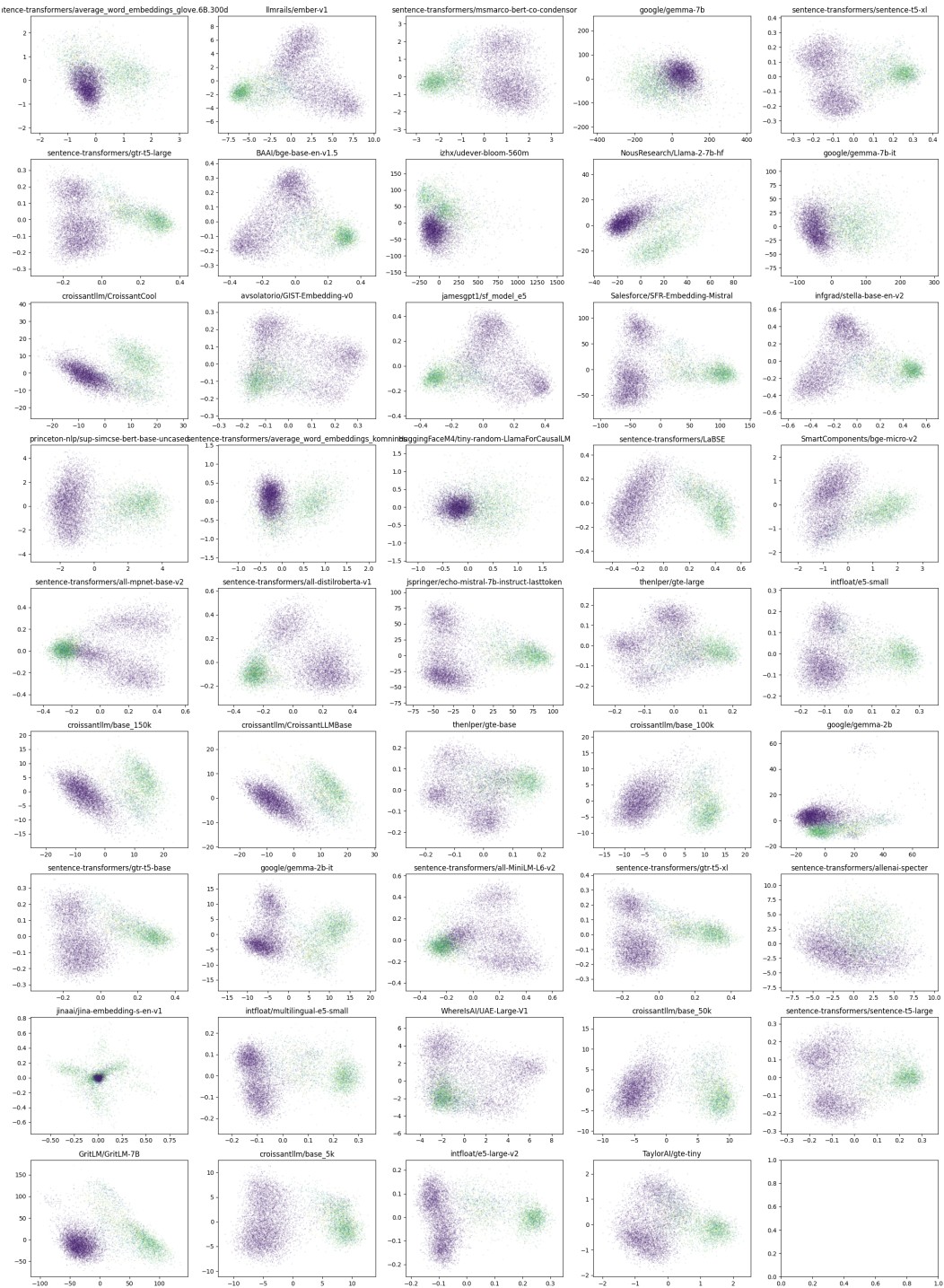

Figure 18: 2D Projection of the embeddings of the models in the first two principal components colored by datasets in NLP

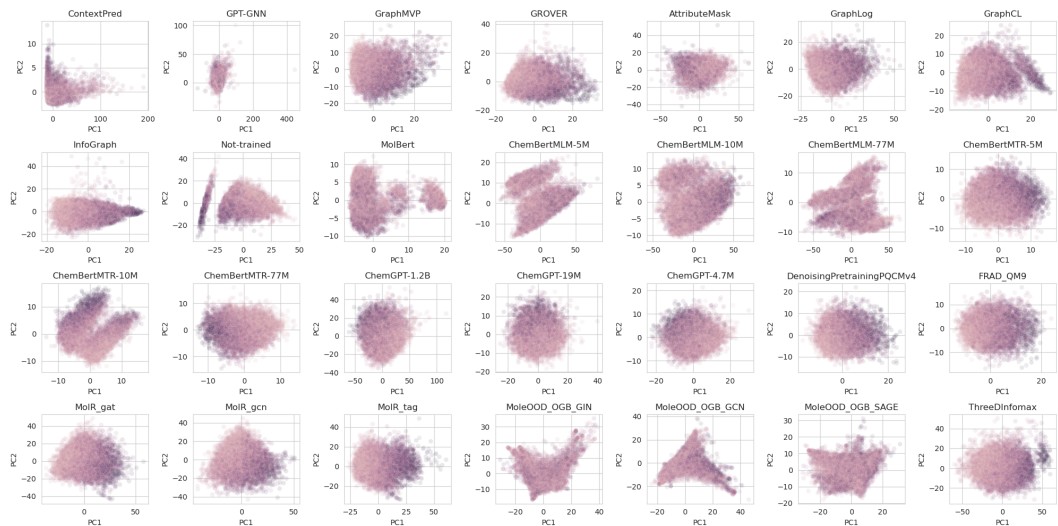

Figure 19: 2D Projection of the embeddings of the models in the first two principal components colored by datasets in molecular modeling. Hue corresponds to the synthetic accessibility score [34].

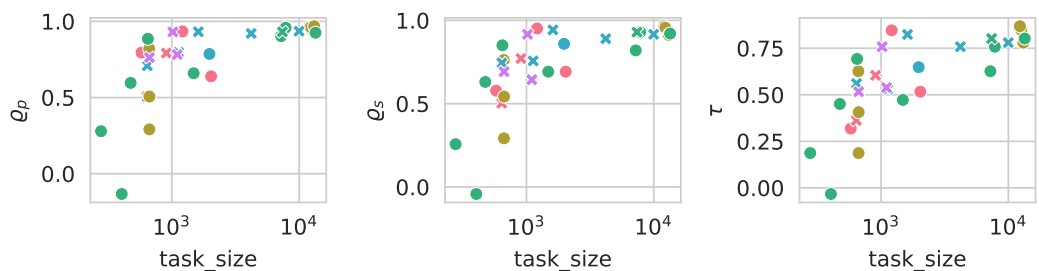

Figure 21: Impact of the task size on the $\overline{\mathcal{I}_S}$ score's ranking's correlation with the downstream tasks performances in molecular modeling in terms of Pearson correlation $\varrho_p$, Spearman correlation $\varrho_s$ and Kendall-Tau $\tau$ coefficient.

### E.4 Impact of the number of models

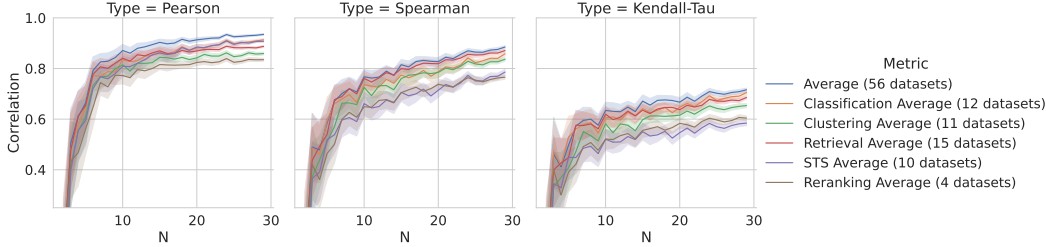

Figure 22: Impact of the number of models used to compute the $\overline{\mathcal{I}_S}$ score in NLP.

We evaluate the strength or $\overline{\mathcal{I}_S}$ score of an embedder with respect to all the others by relaxing the "for all" conditions with the median $\overline{\mathcal{I}_S}$ score. Thus, the number of available embedders might impact the performance of our method. Indeed, if too few embedders are available, it is likely that our evaluation would be biased by favoring models similar to the few available ones. We evaluate the impact of the number of available models by sampling subsets of our global model pool to compute the $\overline{\mathcal{I}_S}$ score. In Figure 22, we found that when fewer models are available, the rankings obtained using the $\overline{\mathcal{I}_S}$

score correlate less with the downstream tasks' performance. However, as the number of models increases, the $\overline{\mathcal{I}_S}$ score becomes a good proxy for the performance of the models on the downstream tasks.

However, the evolution of this correlation is different in the two studied domains. Even with very few models, the $\overline{\mathcal{I}_S}$ score of molecular models already achieves a Spearman correlation close to $0.8$ with the downstream tasks' performance. The correlation is much lower in NLP and only reaches $0.8$ when about 15 models are available. This result can be expected, seeing how sparse the pairwise comparison matrix is in NLP compared to molecular modeling. This matrix is less sparse on molecular data, the graph induced by this adjacency matrix is more connected, and having access to a few nodes is enough to obtain measurements on the whole graph. On the contrary, by being much sparser,

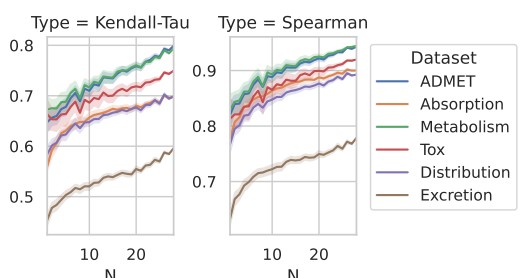

Figure 23: Impact of the number of models used to compute the $\overline{\mathcal{I}_S}$ score in molecular modeling.

having access to a few nodes in this graph in NLP might only give information about the local structure of the graph in the community of the few nodes available. This might explain why the $\overline{\mathcal{I}_S}$ score's correlation with the downstream tasks performance stabilizes in NLP at about ten models, which is equal to the number of communities identified in the graph (in Sec. 5.2).

### E.5 Comparison with other metrics

We evaluate other ways to measure the informativeness (Tab. 9 and Tab. 10) of the embedders. We considered the size of the model, their embeding dimensions and the simple reconstruction error when we train a cross-encoder to go from one embedding to another. Overall we found that our information sufficiency score significantly outperformed these naive metrics in both modalities.

In particular, training a cross-encoder with the $l_2$ reconstruction error proves to correlate with the ability to enable downstream performance, however, this correlation remains weaker compared to using the information sufficiency.

Table 9: Comparison with Baselines: Size of the Embedder, Dimension of the embedding output ($d$) and the $\ell_2$ reconstruction error of the embeddings for NLP datasets.

| | Size | | | d | | | $\bar{I}_{\mathbf{S}}$ | | | $\bar{\ell}_2$ | | |
|---|---|---|---|---|---|---|---|---|---|---|---|---|
| | $\varrho_p$ | $\varrho_s$ | $\tau$ | $\varrho_p$ | $\varrho_s$ | $\tau$ | $\varrho_p$ | $\varrho_s$ | $\tau$ | $\varrho_p$ | $\varrho_s$ | $\tau$ |
| Classification (12 datasets) | 0.46 | 0.42 | 0.32 | 0.52 | 0.66 | 0.55 | **0.92** | **0.88** | **0.73** | -0.79 | -0.85 | -0.66 |
| Retrieval (15 datasets) | 0.40 | 0.39 | 0.29 | 0.46 | 0.63 | 0.52 | **0.89** | **0.89** | **0.70** | -0.71 | -0.84 | -0.65 |
| Clustering (11 datasets) | 0.45 | 0.38 | 0.26 | 0.54 | 0.67 | 0.55 | **0.86** | **0.85** | **0.67** | -0.80 | -0.84 | -0.66 |
| STS (10 datasets) | 0.27 | 0.35 | 0.25 | 0.34 | 0.66 | 0.52 | **0.92** | **0.82** | **0.62** | -0.70 | -0.83 | -0.64 |
| Reranking (4 datasets) | 0.33 | 0.33 | 0.26 | 0.41 | 0.61 | 0.50 | **0.84** | **0.79** | **0.64** | -0.71 | -0.78 | -0.59 |
| Average (56 datasets) | 0.41 | 0.41 | 0.31 | 0.48 | 0.62 | 0.50 | **0.94** | **0.90** | **0.74** | -0.77 | -0.84 | -0.65 |
| Additional Classif (8 datasets) | 0.41 | 0.62 | 0.47 | 0.43 | 0.64 | 0.55 | **0.89** | **0.84** | **0.66** | -0.65 | -0.72 | -0.55 |

Table 10: Comparison with Baselines: Size of the Embedder, Dimension of the embedding output ($d$) and the $\ell_2$ reconstruction error of the embeddings for Molecular Modeling datasets.

| | Size | | | d | | | $\bar{I}_{\mathbf{S}}$ | | | $\bar{\ell}_2$ | | |
|---|---|---|---|---|---|---|---|---|---|---|---|---|
| | $\varrho_p$ | $\varrho_s$ | $\tau$ | $\varrho_p$ | $\varrho_s$ | $\tau$ | $\varrho_p$ | $\varrho_s$ | $\tau$ | $\varrho_p$ | $\varrho_s$ | $\tau$ |
| **A**bsorption (8 datasets) | - | -0.21 | -0.16 | - | -0.43 | -0.29 | - | **0.89** | **0.70** | - | **-0.89** | **-0.70** |
| **D**istribution (3 datasets) | - | -0.07 | -0.03 | - | -0.46 | -0.31 | - | **0.89** | **0.70** | - | -0.86 | -0.66 |
| **M**etabolism (8 datasets) | - | 0.06 | 0.03 | - | -0.46 | -0.34 | - | **0.94** | **0.79** | - | -0.90 | -0.71 |
| **E**xcretion (3 datasets) | - | -0.17 | -0.11 | - | -0.24 | -0.18 | - | **0.77** | **0.60** | - | **-0.77** | -0.56 |
| **T**oxicity (9 datasets) | - | 0.09 | 0.06 | - | -0.49 | -0.35 | - | **0.92** | **0.75** | - | -0.86 | -0.67 |
| **ADMET** (31 datasets) | - | -0.01 | 0.01 | - | -0.47 | -0.32 | - | **0.94** | **0.80** | - | -0.90 | -0.72 |

### E.6 Computational ressources

Evaluating the $\overline{\overline{\mathcal{I}_S}}$ score of the models is computationally inexpensive. Evaluating the $\overline{\overline{\mathcal{I}_S}}$ score requires only a single (small) GPU. All our experiments were conducted on NVIDIA V100 and NVIDIA A6000 GPUs.

Our method's main (computational) shortcoming stems from the need to compute the information sufficiency between all pairs of models. This is a quadratic operation in the number of models. However, in practice, optimizing and estimating the information sufficiency presented in Sec. 3 is cheap. The complete evaluation of the $45$ NLP models can be done in less than $6$ hours on a single GPU.

