# OpenReview forum: "When is an Embedding Model  More Promising than Another?"
_NeurIPS.cc/2024/Conference — NeurIPS 2024 poster_

### Official Review · Reviewer_4AaU · 2024-07-09

**Soundness:** 3
**Presentation:** 4
**Contribution:** 3
**Rating:** 7
**Confidence:** 4

**Summary:**

The authors propose a task-agnostic method for the evaluation of embedding models called "information sufficiency". The general notion is to generate a pairwise matrix that effectively measures how well each embedding model can be used to generate the information content of the other. They compute an overall "information sufficiency score" for each model by computing the median of this pairwise score along one axis. They demonstrate that their metric correlates well with overall downstream task performance.

**Strengths:**

The paper is very well-put-together and cleanly written. The intuition behind the method is well-motivated and makes sense. Most of my outstanding questions were addressed in the supplement. The topic of comparing embedding models is of substantial relevance and interest. There is a need for a better understanding of the quality of embedding models and for a task-agnostic evaluation metric.

**Weaknesses:**

My primary concern centers on the potential pitfalls from using this method, described in more detail in the questions below. I am very open to raising my current score of 6 if those concerns can be addressed sufficiently.

**Questions:**

The metric proposed here is a pairwise metric where the embedding model that can most effectively simulate other embedding models is ranked as the most effective. I wonder if it is possible that if a particular embedding model contains unique information that is not represented by any other embedding model in the chosen set, then is it possible that said embedding model would be ranked poorly? The thrust of my concern is that the adoption of this metric would favor embeddings that are more "central" even in edge cases where more niche information might be more useful.

As a follow up to the last question, I am wondering in what sense the question of whether one embedding model is better than another is generally useful. For most applications and downstream tasks, one does not want an embedding model that is better on average for a host of applications, they want one that is better for their proscribed use case, whatever that may be. Of course, there is the challenge that it is impractical to "try every model", given the breadth of available models. Is there some middle ground here, where a similar method can be used to narrow the number of models that are tested rather than to recommend a single one?

The authors acknowledge that the usefulness of their methods depends on sampling a large number of diverse embedders with which to compare. To what degree is this circular - does this constraint require that you evaluate the set of embedders using a more traditional downstream benchmarking procedure?

The underlying method has some real similarity to the task affinity matrix from the "Vision Taskonomy" work by Zamir et al. (which is currently not cited), although there the idea is not so much to better compare embedding models as it is to understand the informatic structure of tasks in vision literature. Is there perhaps a way of using this pairwise similarity matrix to embed the differences in informatic content across these embedding models using this related approach?


Minor aesthetic/typographical comments:

There is the floating phrase "inforatmation sufficiency" on page 26 in the supplement.

Holy Boldface Batman. That's a lot of boldface. Especially the sentences that start as normal and lead into boldface.

**Limitations:**

Please see my questions. No obvious negative societal impacts of the work.

---

> ### Author Rebuttal · Authors · 2024-08-05
>
> We warmly thank Reviewer 4AaU for their detailed review and the effort they put into evaluating our work.
>
> **Questions.**
>
> 1. **Orthogonal models and unique information.** Yes, a model could contain unique information and still be competitive while not predicting the others.  Our method allows the discovery of such situations by studying the communities in the predictiveness graph: a model containing only unique information would appear as a disconnected component, which should prompt further investigation. A larger set and a more diverse set of models naturally reduce this risk. (See Sec E.4 for a study of the robustness of the method for different sampling of reference models Figure 22 and Figure 23). In molecular modeling, both GPT-GNN and ContextPred appear unique and disconnected, but both of these models also end up being the worst-performing models. In our experiments, we found that models containing only niche information did not perform well, suggesting that good models have to contain, to some extent, a minimal amount of common knowledge.
>
>
> 2. **Evaluation of embedding models.** Our method aims to evaluate generalistic embedders that could be used for a host of possible tasks for a data distribution of interest, thus evaluating foundational embedding models. As you pointed out, it is useful to identify interesting candidates to be used as starting points to evaluate some downstream tasks, but also, more generally, one might not want to rely on different embedders for different tasks because of the potential computational overhead and be interested in the more “generally good” embedder for their data distribution of interest. Pursuing these goals, our method provides the ranking of embedders and community analysis, uncovering similar embedders and orthogonal communities that might bear different types of information. An interesting direction would be to explore if the different clusters discovered by our method perform differently on different tasks, supporting the hypothesis that they capture different types of information (we provided a short analysis of this question in sec C.3.5 and sec D.3).
>
> 3. **Diversity of reference models.** We considered the diversity of architectures, training distribution, and training objectives, and we suppose that these differences lead to different information being captured. We do not rely on performances on downstream tasks (which would indeed be circular), in Sec. E.4, Figures 22 and 23, we study the robustness of our method to the sampling of reference models and the number of sampled models.  \
> In NLP, 10 models suffice to reliably achieve strong correlations (~ 0.90) between the informativeness score and the performances of the model, and in molecular modeling, while with very few models, the correlations are still high (above 0.8), with 10 models we reliably achieve a correlation above 0.9. We computed the correlation scores for different subsets of reference models and reported the average and standard deviation of the correlation score. We will include this discussion in the body of the paper.
> 4. **Vision Taskonomy.** It is indeed very interesting work that we shall include in our related work, and it also appeared extremely relevant to the follow-up work we were conducting! \
> However, it remains unclear to us how to directly connect it to this current work. We identify two possible directions.
>     1. Frame the objectives of comparing embedders using their proposed methodology by adapting the notion of tasks in the paper of Zamir et al.
>     2. Apply the ordinal normalization and the taxonomy discovery procedure to the predictiveness matrix we get using the information sufficiency.
>
>     	Both approaches are non-trivial to execute as it is unclear how to reproduce the transfer modeling for each models/tasks (especially considering the different modalities used) and what would be the optimization constraints for the taxonomy discovery (they are looking for the taxonomy that minimizes the overall amount of training data required to train all models to produce a “scheduling of training”).
>
>
>     Pertaining to your question, we ran eigenvalues analysis and spectral embedding of the embedders based on the similarity matrix to extract communities and evaluate embedders' proximities and found similar results as those reported in the community analysis presented in our work (See Figure 1, in the PDF page attached to the general comment). We will include those in the appendices.
>
>
> We extend our thanks to Reviewer 4AaU for their thorough review of our work, and we hope our answers address all their concerns so they can reassess our work’s quality.

---

> > ### Comment · Reviewer_4AaU · 2024-08-07
> > **Raising my score**
> >
> > Thank you to the authors for their rebuttal answers. I believe my concerns have mostly been addressed. I am raising my score to a 7 to reflect this. Regarding my mention of the Zamir work, my referral was merely prompted by the fact that the high level idea is simply very similar (essentially, building an embedding for a representation out of profile of transfer properties), and was not meant to request any particular analysis for you to pursue.

---

### Official Review · Reviewer_UcaH · 2024-07-12

**Soundness:** 3
**Presentation:** 4
**Contribution:** 3
**Rating:** 7
**Confidence:** 3

**Summary:**

This article introduces a novel framework, grounded in information theory, for assessing the relevance of embedding models (*embedders*). The authors begin by introducing the notion of *sufficiency* of a model A relatively to a model B, which can be used to rank embedders. They prove:
1- 	sufficiency implies (i) a higher capacity to distinguish concepts (Prop. 1)
2- 	sufficiency is equivalent to (ii) having a lower expected Bayes risk (Prop. 2)
They reasonably assume that (i) and (ii) is equivalent to “being a good embedder”.
The authors then introduce the *deficiency* between two models, a relaxed version of sufficiency. Theoretical bounds on the Bayes Risk, as a function of the deficiency, are given (Corr. 1). As deficiency is hard to compute, they develop a surrogate estimator termed *information efficiency* (Definition 3), which can be empirically estimated.

The authors perform experiments to validate their theoretical findings using datasets from NLP and molecular modelling. In the NLP domain, they employ datasets from the MTEB benchmark to demonstrate the correlation between their new ranking scores and model performance. For molecular modelling, they use the ADMET dataset. The authors also propose a graph visualization of the models build from the pairwise information efficiency values.

**Strengths:**

- In this paper, the authors propose an original task, and the approach developed is new.
- The paper is well organized; theory and experiments are well motivated. The progression of the authors' reasoning is very clear.
- The proposed method allows for a graph representation of embedders according to their relative expected performance on the considered dataset.

**Weaknesses:**

- Several key points of the method are not explicit:
    - While the main metric is well described, there is little information on how it is estimated in practice. Besides the fact the KNIFE estimator from [82] is used to quantify mutual information, no procedure is described and the reader is left to try to rebuild it.
    - How to choose reference data is not elaborated on, besides L229-230.
    - The choice the median, discussion about number of models, or embedding dimensions are not discussed within the core of the article while it would help a lot to grasp intuitions about how the proposed metric behaves.
- On motivation and applicability:
     - It would be interesting for the authors to expand on the practical utility of such a method. While it seems to be an interesting alternative to benchmarks which does not require any labelled data, I am not sure of how interesting it could be for applications, especially considering the paper does not expand on how difficult and computationally costly the estimation procedure is.

**Questions:**

- You mention (L211-212) that “We hence attempt to simulate $Z$ from $U$ by learning a Markov kernel”: do you need to learn a Markov Kernel for each model ? Is that computationally expensive ?
- While framing the comparison between embedding models this way is new, it would be interesting to have . However, it would be interesting to have a baseline, even the naivest one (e.g. the number of parameters maybe?) to be able to compare your results.

**Limitations:**

Limitations have been adequately addressed.

---

> ### Author Rebuttal · Authors · 2024-08-06
>
> We would like to thank Reviewer UcaH for their detailed account of our work and the efforts they put into their review.
>
> **Weaknesses.**
>
> 1. **Estimation of $I_S(U \longrightarrow Z)$ (See PDF in general comment).** For a given dataset $D$, we generate the embeddings $(u_i, z_i)$. We then fit a Gaussian mixture on the embeddings $(z_i)_i$ and a parametric Gaussian mixture on the embeddings $(z_i)_i$, parametrized by $(u_i)_i$, i.e. the means and covariance matrices are estimated by a small feed-forward network taking $u_i$ as input, minimizing the negative log-likelihood of the samples $(z_i)_i$. We then compute the corresponding entropies using those trained distributions (see additional PDF, for the detailed algorithm, we will add this algorithm alongside the detailed description to the core of the paper using the additional page allowed to address the reviewers’ comments). In addition, the code source to perform this estimation is available as part of the supplementary material submitted alongside the paper.
> 2. **Choice of Reference Data**. We focused on global evaluation with a large and diverse set of reference data that would be representative of the data distribution of most of the evaluated downstream tasks (ZINC dataset in drug discovery). In Section C.3.6, Figure 10, we used different reference datasets to evaluate our metric. We found that evaluating on data that are close to the ones of the downstream tasks produces better correlation. The most striking example is the performance on the IMBD classification task (evaluating the sentiment of a film review). When our metric is evaluated on the AG News dataset (a dataset of news articles) the correlation with that task is significantly lower than when evaluated with the overall common set or with the amazon polarity dataset, which consists of amazon reviews that are arguably close to film reviews. We plan to extend our work to practical settings to evaluate models on different reference sets to find the best embedder for a given modality or subdistribution.
> 3. **Implementation details in the paper body.** We will use the additional page allowed to include reviewers’ comments to include details on the $I_S$ estimation and discuss our different choices  (number of models, choice of the median, and embedding dimensions) to give more intuitions to the reader.
> 4. **Practical utility.** The main use case for our method is to compare different foundation models as it focuses on how well data are separated in the embedding space. In NLP, foundation models are trained on vast amounts of very diverse textual data, but it is unclear if it embeds well certain types of data (longer texts, simple reviews or question/answer passages etc…). In molecular modeling, since tasks are often extracted from wet-lab experiments, obtaining these labels is expensive, time-consuming, and they are often noisy. In molecular embeddings, the Information sufficiency graph and the identification of communities is helpful to see how the information encoded by a 3d model is inaccessible to the 2D models so far, even though these 2d models were trained to incorporate this information (see Sec D.2). We are currently working on applications of this work as follow up work to evaluate foundational models in medical computer vision: the goal is to evaluate the quality of the embedding models for the different modalities and data distribution they are supposed to handle.
>
> 5. **Computational costs.** We discussed the computational cost of our method in Sec. E.5. Our method is actually surprisingly cheap computationally. Computing the informativeness of a model requires computing $N$ information sufficiency, where $N$ is the number of reference models, each requires fitting two mixtures of Gaussians (the marginal and the conditional). In practice, if the embeddings are precomputed (~ $150$k samples per embedder), it took us less than $6$hours on a single GPU to compute the $45 \times 45$ (all the pairs) information sufficiency to evaluate all the text embedders. For reference, evaluating the whole MTEB benchmark for a model takes hours.
>
> **Questions.**
>
> 1. **Learning kernels.** Yes, we need to fit a kernel per pair of embedders. It grows quadratically in terms of the number of embedders to build the whole graph. However, the kernels are small (3 layers feedforward networks to evaluate the conditional distributions), and the overall evaluation is quick, as discussed in our computational costs analysis (Sec E.5).
> 2. **Baselines.** We included 3 additional baselines: the model sizes (as you suggested), the embedding dimensions, and a less naive one: the reconstruction error of a cross-encoder, a simple feedforward network trained to transform an embedding into another directly, and we use the average l2 loss as a score. Our method consistently outperforms all baselines, reaching higher correlation scores on all considered benchmarks (using the l2 score, the Spearman correlation achieves -0.84 and -0.9 in NLP and molecular modeling, respectively, compared to 0.9 and 0.94 with our method; more details are provided in the general comment).
>
> We renew our thanks to Reviewer UcaH for their thorough review of our work, and we hope our answers addressed all their concerns so they can re-assess our work’s quality.

---

> > ### Comment · Reviewer_UcaH · 2024-08-13
> >
> > I would like to thank the authors for their rebuttal. I find it (along with rebuttals to other reviewers) has clarified things for me. Given this, and the proposed changes, I have updated my rating and recommend acceptance.

---

### Official Review · Reviewer_o2cb · 2024-07-13

**Soundness:** 2
**Presentation:** 3
**Contribution:** 2
**Rating:** 7
**Confidence:** 4

**Summary:**

Evaluating embedding models is challenging because it typically relies on various downstream task data, despite the embedding models being trained for general purposes. This paper introduces an information-theoretic metric for comparing embedding models, eliminating the need for labeled datasets in their evaluation. The core idea is that if one embedding model can simulate another in most cases, it indicates a higher capacity for distinguishing concepts by its embeddings. Empirical results in both text and molecule embedding models show that the proposed metric closely aligns with downstream task rankings. Additionally, the community analysis facilitated by this metric reveals clusters of embedding models, illustrating their relationships.

**Strengths:**

- The target problem is clear, and the proposed metric is well-motivated from information theory perspective.
- The experiments are extensive, demonstrating the effectiveness of the proposed metric.
- The method allows for community analysis of embedding models, which is quite interesting.
- The paper is well-written and easy to follow.

**Weaknesses:**

- The connection between the concept of deficiency and information sufficiency is somewhat unclear.
- The computation of information sufficiency in practice is not well-explained. Adding more details would be beneficial.
- (Minor) The arrangement of tables and figures could be improved.

**Questions:**

- Does transitivity (or a similar weaker concept) hold for information sufficiency? For example, if $I_S(U → Z) > I_S(W → Z)$ and $I_S(W → Z) > I_S(X → Z),$ does $I_S(U → Z) > I_S(X → Z)$?
- Are there specific reasons for using multivariate Gaussians in Markov kernel learning? Are there any alternatives?
- What are the requirements for data in estimating information sufficiency? How do the quantity, quality, and diversity of datasets affect the effectiveness of the proposed metric?
- Line 114: Should $P_{U|X}$ be $P_{Z|X}$?
- Lines 188-189: Isn't this bound practically vacuous due to the size of X?

**Limitations:**

The authors adequately address the limitations.

---

> ### Author Rebuttal · Authors · 2024-08-06
>
> We thank Reviewer o2cb for their review and the interesting questions they raised. We do our best to answer them below.
>
> **Weaknesses**
>
> 1. **Information sufficiency and deficiency. (See Official comment below for the proof)** We can establish the following connection between deficiency and mutual information, where information sufficiency serves as an estimate:
> $$
>  \\delta(P_{U|X} \rightarrow P_{Z|X} ) \ge\inf_M \\mathbf{E}[\ell(X,M)] \\ge R_{X,\ell}^{-1}(I(Z,X;U))
> $$
> Where $R_{X,\ell}$ is the rate-distortion function for $X$ and a loss function $\ell(\cdot)$. **This is a non-increasing function in mutual information: when mutual information is low, the deficiency is high. This suggests that the mutual information $I(Z;U)$ should be as large as possible as a necessary condition to achieve lower deficiency.**
> __Due to character limits, the proof is provided in the following comment. We will include it in the final version of the paper.__
>
> 2. **Estimation of $I_S(U \longrightarrow Z)$.** For a given dataset $D$, we generate the embeddings $(u_i, z_i)$. We then fit a Gaussian mixture on the embeddings $(z_i)_i$ and a parametric Gaussian mixture on the embeddings $(z_i)_i$, parametrized by $(u_i)_i$, i.e. the means and covariance matrices are estimated by a small feed-forward network taking $u_i$ as input, minimizing the negative log-likelihood of the samples $(z_i)_i$. We then compute the corresponding entropies using those trained distributions. (See additional PDF, for the detailed algorithm, we will add this algorithm alongside the detailed description to the core of the paper using the additional page allowed to address the reviewers’ comments). In addition, the code source to perform this estimation is available as part of the supplementary material submitted alongside the paper.
> 3. We will further improve the pagination and organization of the tables and figures.
>
> **Questions**
>
> 1. **Transitivity.** The example provided in the review: If $I_s (U→Z)>=I_s (W→Z)$ and $I_s (W→Z)>= I_s (X→Z)$, then $I_s (U→Z)>=I_s (X→Z)$ is always true by the transitivity of the “>=” relationship as boils down to $(a \geq b \text{ and } b \geq c) \implies a\geq c$.
> **A less straightforward transitivity property exists for the deficiency**, indeed we can show that
> $\delta(P_{A|X}\rightarrow P_{C|X}) \leq \delta(P_{A|X}\rightarrow P_{B|X}) + \delta(P_{B|X}\rightarrow P_{C|X})$.  This relation simply states that the reconstruction error of an embedder C by an embedder A is lower than the reconstruction of any embedder B by A, and the embedder C by B. __We include the proof at the end of this rebuttal and will add it to the final version of the paper__
>
> 1. **Choice of Gaussian Mixtures.** Gaussian mixtures are known to be universal estimators of densities, and to our knowledge, the KNIFE paper corresponds to the state of the art for this kind of information-theoretic estimations. Another option is to directly fit cross-encoders (feedforward networks) using the MSE loss and use this reconstruction loss as score of informativeness. Our method consistently outperforms this reconstruction loss strategy, reaching higher correlation scores (a Spearman correlation of 0.84 compared 0.9 in NLP and 0.9 compared to 0.94 in molecular modeling on all benchmarks) on every benchmark we considered (See attached PDF and general comment for full results and additional baselines).
> 2. **Reference set requirements**. The requirement is that the dataset used is sufficiently large to correctly represent the distribution of data that will be presented in practice (the ZINC dataset for molecules for example). In Section C.3.6, Figure 10, we evaluate the impact of evaluating the I_S on different subsets of data on the correlation with the downstream tasks performance. Evaluating broader data tends to help, but the performance is even stronger if the data used to evaluate the I_S are close to the data used for the downstream tasks. For example, evaluating $I_S$ on the Amazon polarity gives good insights on the performance on the imdb dataset whose task is related: the goal is to evaluate the sentiment about a movie.
> 3. Yes, this is correct. We will correct this typo.
> 4. **Tightness of the bound.** It might be the case, but we are not interested in actually estimating the bound but rather in the relationships between the terms of said bound since our goal is to compare embedding models. Our goal was to highlight the connection between the deficiency and the different risks: controlling and comparing the deficiency is still a good idea even if the bound is not tight.
>
> We extend our thanks to Reviewer o2cb for their thorough review of our work, and we hope our answers address all their concerns so they can reassess our work’s quality.
>
> ---
>
> __Proof: Transitivity of the deficiency__
> It is easy to check that
> Eq 1:
> $$
>  \\delta(P_{A|X}\\rightarrow P_{C|X})  =:  \\inf_{M\\in \\mathcal{K}(C|A)} \\| M\\!\\circ\\!P_{A|X} -  P_{C|X} \\|\_{\\text{TV}}  \\leq \\inf_{M\\in \\mathcal{K}(C|B)} \\inf_{ M'\\in \\mathcal{K}(B|A)} \\| M\\!\\circ (M'\\!\\circ\\!P_{A|X}) - P_{C|X} \\|_{\\text{TV}}.
> $$
>
> On the other hand, for any Markov kernels $M\in \mathcal{K}(C|B), M'\in \mathcal{K}(B|A)$:
>
> Eq 2:
> $$
> \\| M\\!\\circ M'\\!\\circ\\!P_{A|X} - P_{C|X} \\|\_{\\text{TV}}= \\| M\\!\\circ M'\\!\\circ\\!P_{A|X} -  M\\!\\circ\\!P_{B|X} +  M\\!\\circ\\!P_{B|X} - P_{C|X} \\|\_{\\text{TV}} \\leq \\| M\\!\\circ M'\\!\\circ\\!P\_{A|X} -  M\\!\\circ\\!P\_{B|X} \\|_{\text{TV}}+\\|  M\\!\\circ\\!P\_{B|X} - P\_{C|X} \\|\_{\\text{TV}},
> $$
>
> and by data processing inequality on the TV norm:
> Eq 3:
> $$
> \\| M\\!\\circ M'\\!\\circ\\!P_{A|X} -  M\\!\\circ\\!P_{B|X} \\|\_{\text{TV}} \\leq \\| M'\\!\\circ\\!P_{A|X} -  P_{B|X} \\|\_{\text{TV}}.
> $$
>
> The desired inequality follows by applying inequality Eq.3 to Eq.2, and then taking the infimum at both sides over all Markov kernels $M\in \mathcal{K}(C|B)$, $M'\in \mathcal{K}(B|A)$, and using inequality Eq.1.

---

> ### Author Response · Authors · 2024-08-07
> **Further details about the deficiency and the mutual information/information sufficiency**
>
> **This is a complement to our answer to Reviewer o2cb regarding the connection between our estimator and the deficiency.**
>
> ## Review of the Distortion-Rate Function
>
> The rate-distortion (RD) function of a random variable --the source-- $X$ for a given distortion function $\ell(\cdot, \cdot)$ is defined as [Cover 2006]
> $$
> R_{X,\\ell}(D) \,\triangleq \inf_{\rule{0mm}{4.3mm}\substack{p_{\widehat{X}|X}:\\ \mathbf{E}[\ell(X,\widehat{X})] \leq D}} I(X;\\widehat{X}).
> $$
>
> Furthermore, we assume that there exist $D>0$ such that $R_{X,\ell}(D)$ is finite.
> We denote the infimum of those $D$ by $D_{\min}$ and $R_{\max}\triangleq R_{X,\ell}(D_{\min})$ (or, more precisely, $R_{\max}\triangleq \lim_{D\rightarrow  D_{\min}+}R(D)$).
> The following properties (see [Lem.~1.1, Csiszar 1974]) of the RD function will be used in what follows.
>
> __Theorem 1:__
> The RD function $R_{X,\ell}(D)$ is a non-increasing convex function of $D$ on the interval $(D_{\min}, \infty)$.
> It is monotonically decreasing on the interval $(D\_{\\min},D\_{\\max})$ and constant with $R_{X,\ell}(D)=R\_{\min}$ on $[D_{\max},\infty)$ (here $D_{\max}=\infty$ and $D_{\min}=0$ are possible). The inverse function $R_{X,\ell}^{-1}(r)$ is well defined on $(R_{\min}, R_{\max})$ and monotonically decreasing.
>
> The inverse function $R_{X,\ell}^{-1}(r)$ is known as the distortion-rate (DR)
> function of the random variable $X$ for the given distortion function $\ell(\cdot, \cdot)$.
>
> ## Deficiency and Information Sufficiency
>
> Assume two Markov (or transition probability) kernel between $\mathsf{U}$ and $\mathsf{Z}$ which is a mapping $M : \mathcal{B}(\mathsf{Z}) \times \mathsf{U} \rightarrow [0,1]$,  and between $\mathsf{Z}$ and $\mathsf{U}$ which is a mapping $M^\prime : \mathcal{B}(\mathsf{U}) \times \mathsf{Z}  \rightarrow  [0,1]$. From the embedder definition, we have the Markov chain  $Z \leftrightarrow X \leftrightarrow U$.
>
> We begin by the deficiency  $\delta(P_{U|X} \rightarrow P_{Z|X} ) $. Let us define a suitable distortion function:
> $$
> \ell(x, M) \triangleq  \\|  M \circ P_{U|X=x} - P_{Z|X=x} \\|_{\text{TV}}.
> $$
>  From which, it is easy to check that
>
> __Equation 1__
> $$
>  \delta(P_{U|X} \rightarrow P_{Z|X} ) \triangleq  \inf_M \\| M \circ P_{U|X} - P\_{Z|X} \\|_{\text{TV}} \geq \inf_M\mathbf{E}[\ell(X,M)],
> $$
> where the last inequality follows by replacing the supremum over $x\in \mathsf{X}$ by the expectation. Using the data processing inequality and the definition of the RD function, we obtain the following bound for any Markov kernel $M$:
>
> __Equation 2__
> $$
> I(Z,X;U)  \geq
> \inf_{\\rule{0mm}{4.3mm}\substack{p_{\widehat{X}|X}:\\ \mathbf{E}[\ell(X,\widehat{X})] \leq \mathbf{E}[\ell(X,M)]}}  I(X;\widehat{X})
> = R(\mathbf{E}[\ell(X,M)]).
> $$
> For $\mathbf{E}[\ell(X,M)]\in (D_{\min},D_{\max})$,
> we can invert the RD function, and thus we obtain from it the fundamental bound $R_{X,\ell}^{-1}(I(Z,X;U))\leq \mathbf{E}[\ell(X,M)]$  or, equivalently,
>
> __Equation 3__:
> $$
>  \delta(P_{U|X} \rightarrow P_{Z|X} ) \geq \inf_M \mathbf{E}[\ell(X,M)] \geq R_{X,\ell}^{-1}(I(Z,X;U)) ,
> $$
> which follows from inequality Eq.1.
> For $\mathbf{E}[\ell(X,M)] < D_{\min}$, Equation 2 reduces to $I(Z,X;U) \geq +\infty$ which shows that in order to achieve an expected distortion below $D_{\min}$ the random variables $(Z,X,U)$ must have a joint distribution that is not absolutely continuous with respect to the product of their maginal distributions $(Z,X)$ and $U$.
> For $\mathbf{E}[\ell(X,M)] \geq D_{\max}$ we obtain the trivial bound $I(Z,X;U) \geq 0$.
> **The lower bound in Equation 3 is a non-increasing function in the mutual information. This suggests that the mutual information $I(Z;U)$ should be as large as possible as a necessary condition to achieve lower deficiency.** Similarly, we can obtain that the symmetric bound:
>
> __Equation 4__
> $$
> \rule{0mm}{5mm}\rule{3mm}{0mm}
>  \delta(P_{Z|X} \rightarrow P_{U|X} ) \geq \inf_{M^\prime} \mathbf{E}[\ell(X,M^\prime)] \geq R_{X,\ell}^{-1}(I(U,X;Z)).
> $$
>
> We will incorporate these suggestions into the final version of the paper.
>
> ## References
> T. M. Cover and J. A. Thomas. Elements of Information Theory. Wiley, New York, NY, 2nd edition, 2006.
>
> I. Csiszar, “On an extremum problem of information theory,” Studia Scientiarum Mathematicarum Hungarica, vol. 9, no. 1, pp. 57–71, 1974.

---

> > ### Comment · Reviewer_o2cb · 2024-08-10
> >
> > Thank you for your detailed response! My questions and concerns are well-addressed and I actually learned a lot. I've raised my rating to 7.

---

### Official Review · Reviewer_id3c · 2024-07-27

**Soundness:** 3
**Presentation:** 4
**Contribution:** 4
**Rating:** 7
**Confidence:** 3

**Summary:**

This paper proposes a new metric to compare embedding models without relying on labeled data. The approach involves embedding data using two separate neural networks Z and U, and then trying to use embedding model U to simulate/match the output of embedding Z. They then use this to calculate an information sufficiency criterion. They motivate this with theory, and show that their information sufficiency metric aligns nicely with established embedding benchmarks across two separate domains, NLP and molecular biology.

**Strengths:**

This is an innovative, statistics-based approach that can potentially add a lot to the rapidly evolving subfield of embedding models.

This paper validates their approach across completely different domains: NLP and molecular biology. The authors have run extensive experiments on tens of embedding models and many different datasets within the two domains.

This paper is very well written with an attention to detail (and an extensive appendix). The paper also does a valiant job of combining theory and experiment.

**Weaknesses:**

1. This paper puts a heavy emphasis on the theoretical motivation without giving enough detail about the actual algorithm for calculating the information sufficiency metric I_s. Line 211 states “We hence attempt to simulate Z from U by learning a Markov kernel M via a mixture of multivariate Gaussians, and measure the uncertainty reduction it induces.” It would be helpful to expand on this in detail. The reader is left only guessing how exactly this was done, and in my opinion it does not seem quite replicable. The details in the Estimation method section E.1 and Hyperparameter selection section E.2 are somewhat minimal.

2. This method is touted as a principled way to compare embedding models without using data labels (but still using data in the desired domain). However it is unclear what the other tradeoffs are. Aside from the question of data labels, Is this approach more computationally efficient?

**Questions:**

1. How exactly do you calculate the information sufficiency metric I_s for a given dataset in MTEB and two models? Can you expand on this in more detail (as mentioned above)?

2. In the discussion section, you mention that there are other potentially promising methods for learning the Markov kernel. What do you think some other promising approaches might be?

3. In Section C.3.1, the authors admit that their results are better for MTEB classification tasks when compared to STS, clustering, and reranking tasks. What numbers/tables are they referring to here?

**Limitations:**

It would be helpful for the authors to share code upon publication so that others can use this metric/approach.

### Minor comments:

Typo Figure 3(b) “Molecular Modelling”
Typo line 213 “embbeders”

---

> ### Author Rebuttal · Authors · 2024-08-05
>
> We warmly thank Reviewer id3c for their reviews.
>
> **Weaknesses:**
>
> 1. We refer the reviewer to Questions below.
> 2. **Computational costs.** Our method is cheap computationally. We quickly discussed the computational cost of evaluating our method in Sec. E.5. Computing the informativeness of a model requires evaluating $N$ information sufficiency, where $N$ is the number of reference models, each requires fitting two mixtures of Gaussians (the marginal and the conditional). With precomputed embeddings (~ $150$k samples per embedder), it took us less than $6$hours on a single GPU to compute the $45 \times 45$ (all the pairs) information sufficiency to evaluate all the text embedders. As a result, in our examples, computing the informativeness score of a model takes less than 15 minutes.
>
> **Questions:**
>
>
>
> 1. **Estimation of $I_S(U \longrightarrow Z)$ (See PDF in general comment).** For a given dataset $D$, we generate the embeddings $(u_i, z_i).$ We then fit a Gaussian mixture on the embeddings $(z_i)_i$ and a parametric Gaussian mixture on the embeddings $(z_i)_i$, parametrized by $(u_i)_i$, i.e. the means and covariance matrices are estimated by a small feed-forward network taking $u_i$ as input, minimizing the negative log-likelihood of the samples $(z_i)_i$. We then compute the corresponding empirical entropies using those trained distributions (See the attached PDF file in general comment for the detailed algorithm; we will add it alongside the detailed description to the core of the paper using the additional page allowed to address the reviewers’ comments). In addition, the code source to perform this estimation is available as part of the supplementary material submitted alongside the paper.
> 2. **Better estimators of the deficiency.** The information sufficiency is admittedly a proxy for the actual deficiency we wanted to evaluate. Our decision was guided by tractability and numerical stability arguments (we expand more on this in our answer to Reviewer o2cb). There is still work to be done to find a proven estimator. For example, we were able to show a more direct connection between the deficiency and the critic loss of a GAN. However, we could not make it work in practice: this more grounded approach is too numerically unstable and leads to very poor results (e.g., the information sufficiency yields correlations of $0.90$, while this more grounded approach yielded at most $0.70$). This opens an interesting line of research for the near future.
> 3. **Performance for classification tasks and others.** In Figure 3, table (a), the Kendall-Tau correlation between our metric and the performance on downstream tasks is significantly higher ($0.73$ ) for the classification tasks than for the other type of tasks ($0.69$ at best and closer to $0.65$ globally). Our interpretation of these results is that classification tasks rely on the training of an additional classifier on top of the embeddings, whereas the other tasks only rely on the dot product between embeddings to make decisions (retrieval, similarity and clustering). Our theoretical results give insight into what is doable when learning the best classifier possible, but they do not guarantee that the dot product / l2 distance in the embedding space captures any useful semantics.
>
> **Source code.** We released the source code for this project: it is available as supplementary material attached to the submission and at [[https://anonymous.4open.science/r/emir-B8D3/Readme.md](https://anonymous.4open.science/r/emir-B8D3/Readme.md)]. A public repository will be shared upon acceptance. Our implementation only relies on pre-computed embeddings and thus can directly be used for any domain as long as they have been dumped. We will make sure to expand the Estimation Method section to provide more details on how we estimate information sufficiency in practice, and we will publicly release the library and code as a practical library to estimate our metric.
>
> We hope we were able to alleviate your concerns and that our answers will allow you to improve your already positive assessment of our work, and we remain available for any additional information.

---

### Author Rebuttal · Authors · 2024-08-05

We appreciate that all the reviewers have recognized our work's novelty, significance, and clarity, as well as its comprehensive empirical analysis.

The reviewers raised 3 main concerns: a lack of details about the estimation procedure, its practical usage, and computational cost, and requested additional baselines.

**Estimation of $I_S(U \longrightarrow Z)$.** For a given dataset $D$, we generate the embeddings $(u_i, z_i)$. We then fit a Gaussian mixture on the embeddings $(z_i)_i$ and a parametric Gaussian mixture on the embeddings $(z_i)_i$, parametrized by $(u_i)_i$, i.e. the means and covariance matrices are estimated by a small feed-forward network taking $u_i$ as input, minimizing the negative log-likelihood of the samples $(z_i)_i$. We then compute the corresponding empirical entropies using these learned distributions **(See algorithm in attached PDF for further details).** We will add this algorithm alongside the detailed description to the core of the paper using the additional page allowed to address the reviewers’ comments.

**Source code.** In addition, the source code is available as part of the supplementary material submitted alongside the paper as well as an anonymous github repo: [[https://anonymous.4open.science/r/emir-B8D3/Readme.md](https://anonymous.4open.science/r/emir-B8D3/Readme.md)] and it will be publicly released upon acceptance as an easy-to-use library to evaluate our metric.

**Computational cost and applicability (Section E.5).** Computing the informativeness of a model requires computing $N$ information sufficiency, where $N$ is the number of reference models, each requires fitting two mixtures of Gaussians (the marginal and the conditional). In practice, if the embeddings are precomputed (~ $150$k samples per embedder), it took us less than $6$hours on a single GPU to compute the $45 \times 45$ (all the pairs) information sufficiency to evaluate all the text embedders. As a result, in our examples, computing the informativeness score of a new model takes less than 15 minutes. For reference, evaluating the whole MTEB benchmark takes hours for a single model.

**Additional baselines.**  As suggested by Reviewer UcaH, we included $3$ baselines: the number of parameters of the models, the dimension of the embeddings, and a reconstruction loss score. We fitted simple feed-forward “cross encoders” from one embedder to another. We use their reconstruction loss as a score. **We found that our method consistently outperforms these $3$ baselines.** We report the full results in the PDF page attached to the rebuttals.


|                             	| | Size | | | d | | |$\bar{I}_{\mathbf{S}}$| | |$\bar{\ell}_{2}$| |
|---------------------------------|----------|----------|--------|----------|----------|--------|----------------|----------------|----------------|----------|----------|--------|
|                             	|$\rho_p$|$\rho_s$|$\tau$|$\rho_p$|$\rho_s$|$\tau$|$\rho_p$   	|$\rho_s$   	|$\tau$     	|$\rho_p$|$\rho_s$|$\tau$|
| Classification (12 datasets)  	| 0.46 	| 0.42 	| 0.32   | 0.52 	| 0.66 	| 0.55   |**0.92**|**0.88**|**0.73**| -0.79	| -0.85	| -0.66  |
| Retrieval (15 datasets)      	| 0.40 	| 0.39 	| 0.29   | 0.46 	| 0.63 	| 0.52   |**0.89**|**0.89**|**0.70**| -0.71	| -0.84	| -0.65  |
| Clustering (11 datasets)     	| 0.45 	| 0.38 	| 0.26   | 0.54 	| 0.67 	| 0.55   |**0.86**|**0.85**|**0.67**| -0.80	| -0.84	| -0.66  |
| STS (10 datasets)            	| 0.27 	| 0.35 	| 0.25   | 0.34 	| 0.66 	| 0.52   |**0.92**|**0.82**|**0.62**| -0.70	| -0.83	| -0.64  |
| Reranking (4 datasets)       	| 0.33 	| 0.33 	| 0.26   | 0.41 	| 0.61 	| 0.50   |**0.84**|**0.79**|**0.64**| -0.71	| -0.78	| -0.59  |
| Average (56 datasets)         	| 0.41 	| 0.41 	| 0.31   | 0.48 	| 0.62 	| 0.50   |**0.94**|**0.90**|**0.74**| -0.77	| -0.84	| -0.65  |
| Additional Classif (8 datasets) | 0.41 	| 0.62 	| 0.47   | 0.43 	| 0.64 	| 0.55   |**0.89**|**0.84**|**0.66**| -0.65	| -0.72	| -0.55  |


*Comparison with Baselines: Size of the Embedder, Dimension of the embedding output ($d$) and the $\ell_2$ reconstruction error of the embeddings for NLP datasets.*




|                             	| | Size | | | d | | |$\bar{I}_{\mathbf{S}}$| | |$\bar{\ell}_{2}$| |
|---------------------------------|----------|----------|--------|----------|----------|--------|----------------|----------------|----------------|----------|----------|--------|
|                             	|$\rho_p$|$\rho_s$|$\tau$|$\rho_p$|$\rho_s$|$\tau$|$\rho_p$   	|$\rho_s$   	|$\tau$     	|$\rho_p$|$\rho_s$|$\tau$|
| Absorption (8 datasets)   | -       	| -0.21   	| -0.16  | -       	| -0.43   	| -0.29  | -       	| **0.89** | **0.70** | -       	| **-0.89** | **-0.70** |
|Distribution (3 datasets) | -       	| -0.07   	| -0.03  | -       	| -0.46   	| -0.31  | -       	| **0.89** | **0.70** | -       	| -0.86      	| -0.66      	|
|Metabolism (8 datasets)   | -       	| 0.06    	| 0.03   | -       	| -0.46   	| -0.34  | -       	| **0.94** | **0.79** | -       	| -0.90      	| -0.71      	|
|Excretion (3 datasets)	| -       	| -0.17   	| -0.11  | -       	| -0.24   	| -0.18  | -       	| **0.77** | **0.60** | -       	| **-0.77** | -0.56      	|
|Toxicity (9 datasets) 	| -       	| 0.09    	| 0.06   | -       	| -0.49   	| -0.35  | -       	| **0.92** | **0.75** | -       	| -0.86      	| -0.67      	|
|ADMET (31 datasets)   	| -       	| -0.01   	| 0.01   | -       	| -0.47   	| -0.32  | -       	| **0.94** | **0.80** | -       	| -0.90      	| -0.72      	|


*Comparison with Baselines: Size of the Embedder, Dimension of the embedding output ($d$) and the $\ell_2$ reconstruction error of the embeddings for Molecular Modelling datasets.*

We hope that these clarifications and additional comparisons address the reviewers' concerns and positively influence their evaluation of our work.

---

### Author Response · Authors · 2024-08-12

Dear Reviewers,

We would like to sincerely thank all the reviewers for their diligent work and the efforts they put into reviewing this paper. We understand that the review and discussion process is highly time-consuming, and we sincerely appreciate Reviewers o2cb, 4AaU, and UcaH for engaging with us.

We also hope our responses alleviated the concerns of Reviewer id3c, and we remain available for any further clarifications until the end of the discussion period.

We will ensure to include all the valuable feedback we got to improve the final version of our work.
Best regards, The Authors of Paper #12245

---

### Decision · Program_Chairs · 2024-09-25

**Decision:**

Accept (poster)

**Comment:**

This paper introduces a novel, information-theoretic framework for assessing the relevance of embedding models. The authors introduce the notion of sufficiency of a model A relatively to a model B (used to rank embedders) and tie this to a related notion of informativeness (information efficiency).

Overall, the proposed approach is quite interesting & novel and a refreshing (theoretical) way to quantify the effectiveness of different embedding models. The authors supplement the theory with good analysis & combine it with experimental results to validate the efficacy of the proposed metric. They also did a very nice job of addressing comments/concerns during the rebuttal phase. All the reviewers agree that it’s a solid paper, no concerns.